# RL-SPH: Learning to Achieve Feasible Solutions for Integer Linear Programs

## Abstract

Primal heuristics play a crucial role in quickly finding feasible solutions for NP-hard integer linear programming (ILP). Although *end-to-end learning*-based primal heuristics (E2EPH) have recently been proposed, they are typically unable to independently generate feasible solutions. To address this challenge, we propose RL-SPH, a novel reinforcement learning-based start primal heuristic capable of independently generating feasible solutions, even for ILP involving non-binary integers. Empirically, RL-SPH rapidly obtains high-quality feasible solutions with a 100% feasibility rate, achieving on average a 44× lower primal gap and a 2.3× lower primal integral compared to existing start primal heuristics.

## 1 Introduction

Combinatorial optimization (CO) involves mathematical optimization that aims to minimize or maximize a specific objective function (Mazyavkina et al., 2021). When both the objective function and the constraints of CO are linear, the problem is referred to as linear programming (LP) (Bengio et al., 2021). Furthermore, if the variables in LP are required to take integer values, it becomes an integer linear programming (ILP) (Bertsimas & Tsitsiklis, 1997). ILP has been widely applied to real-world scenarios such as logistics (Kweon et al., 2024), vehicle routing problem (Toth & Vigo, 2002), and path planning (Zuo et al., 2020).

Since ILP is NP-hard, heuristic approaches have attracted significant attention (Berthold, 2006). Primal heuristics aim to quickly find feasible solutions without guaranteeing feasibility (Berthold, 2006; Shoja & Axehill, 2023), in contrast to methods that aim for optimality (Cantürk et al., 2024). Traditional primal heuristics rely on expert knowledge, requiring significant manual effort (Bengio et al., 2021). Recently, ML-based primal heuristics have been proposed (Nair et al., 2020; Shen et al., 2021; Yoon, 2022; Han et al., 2023; Cantürk et al., 2024; Huang et al., 2024; Liu et al., 2025), which fall under the category of *end-to-end learning* (Bengio et al., 2021; Han et al., 2023), as they learn common patterns across ILP instances and directly generate solutions. Figure 1(a) illustrates how existing *end-to-end learning*-based primal heuristics (E2EPH) combined with ILP solvers generate feasible solutions. A trained ML model generates *a partial solution* over integer variables, which is then passed to an ILP solver (e.g., Gurobi and SCIP) to obtain a feasible solution for the subproblem.

E2EPH combined with an ILP solver have shown the ability to efficiently find high-quality solutions by reducing the search space. Despite these advances, ensuring feasibility remains a major challenge, since inaccurate ML predictions can lead to constraint violations (Han et al., 2023; Liu et al., 2025), which pose a significant obstacle to solving ILP. Recent studies (Han et al., 2023; Huang et al., 2024; Liu et al., 2025) have sought to mitigate this risk by adopting trust regions instead of strictly fixing variables (Nair et al., 2020; Yoon, 2022). However, they rely on ILP solvers to obtain feasible solutions. Recently, several E2EPH have been proposed to eliminate the reliance (Zeng et al., 2024; Geng et al., 2025), but they still struggle to obtain them independently. This limitation underscores the need for a new class of E2EPH that can independently produce feasible solutions, known as *start primal heuristics*, which attempt to convert an infeasible solution into a feasible one (Berthold, 2006).

The infeasibility caused by inaccurate ML predictions can be more pronounced for non-binary integer (hereafter, integer) variables due to their wider value range compared to binary variables. Many real-world problems involve integer variables (e.g., logistics (Kweon et al., 2024) and maritime transportation (Papageorgiou et al., 2014)). However, existing E2EPH studies have primarily focused on binary variables (Han et al., 2023; Huang et al., 2024; Liu et al., 2025; Zeng et al., 2024; Geng et al., 2025), highlighting the need for E2EPH capable of effectively handling integer variables.

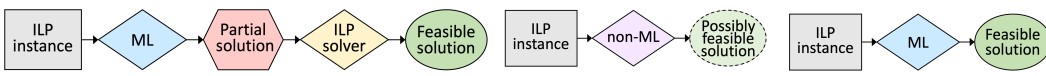

(a) Existing E2EPH with ILP solvers.  (b) Existing SPH.  (c) RL-SPH (Ours).

Figure 1: Comparison among E2E primal heuristics, start primal heuristics (SPH), and our approach.

This work aims to achieve feasibility for ILP, which serves as a crucial prerequisite for achieving optimality. Given that reaching optimality remains a major challenge in the ML for CO community (Son et al., 2023), ensuring feasibility becomes an essential first step in solving ILP. Despite its importance, feasibility is not typically guaranteed by primal heuristics for ILP (Shoja & Axehill, 2023), including the E2EPH methods. To tackle a more challenging setting than prior work—where ML must independently generate feasible solutions for ILP—we propose a novel **R**einforcement **L**earning-based **S**tart **P**rimal **H**euristic, called **RL-SPH**. Figure 1(c) illustrates how RL-SPH generates feasible solutions for ILP. We design a reinforcement learning framework tailored to ILP, which enables the agent to learn variable-constraint relationships and achieve feasibility. RL-SPH operates without the need for near-optimal solutions as training labels, which typically take an hour to obtain per instance (Han et al., 2023; Huang et al., 2024). Instead of directly predicting variable values, RL-SPH learns to decide whether to change the value of each variable. The reward functions are designed to guide the RL agent based on the degree of constraint violations and the quality of the solution. To capture long-range variable dependencies, we adopt a Transformer-based GNN architecture.

Our main contributions are as follows, and Appendix G further clarifies the context that positions these contributions.

- We propose a novel RL-based start primal heuristic, RL-SPH, which learns to independently generate high-quality feasible solutions for ILP. To the best of our knowledge, RL-SPH is the first E2EPH that provides a theoretical feasibility guarantee for ILP (Proposition 1).
- We prove that RL-SPH attains a feasible solution for ILP within polynomial time when the condition $\mathcal{T} \leq n$ is satisfied (Proposition 2).
- We design a feasibility-aware search strategy that leverages the proposed GNN trained via our RL framework tailored to ILP (Algorithm 1). By narrowing the search space using this strategy, the GNN agent efficiently learns variable–constraint and inter-variable relationships through strong reward signals, acquiring effective problem-solving capabilities.
- We demonstrate that RL-SPH outperforms widely used start primal heuristics across four CO benchmarks by achieving 100% feasibility, produces high-quality solutions more efficiently when combined with an ILP solver even on instances with integer variables, and further generalizes its problem-solving capability to unseen MIPLIB instances when jointly trained.

## 2 PRELIMINARIES

### 2.1 INTEGER LINEAR PROGRAMMING

Integer linear programming (ILP) is an optimization problem that minimizes or maximizes a linear objective function, while satisfying linear constraints and integrality constraints on decision variables (Bertsimas & Tsitsiklis, 1997). A standard form of an ILP instance can be formulated as follows:

$$\text{minimize } \mathbf{c}^\top \mathbf{x} \tag{1a}$$

$$\text{subject to } \mathbf{Ax} \leq \mathbf{b} \tag{1b}$$

$$x_i \in \mathbb{Z}, \quad \forall i \tag{1c}$$

$$l_i \leq x_i \leq u_i, \quad \forall i \tag{1d}$$

where $\mathbf{x} \in \mathbb{R}^n$ is a column vector of $n$ decision variables, $\mathbf{c} \in \mathbb{R}^n$ is a column vector of the objective coefficients, $\mathbf{A} \in \mathbb{R}^{m \times n}$ is the constraint coefficient matrix, $\mathbf{b} \in \mathbb{R}^m$ is a column vector of the right-hand side of the constraints, and $l_i/u_i$ denote the lower/upper bounds for each decision variable $x_i$. A solution $\mathbf{x}$ is said to be *feasible* if it satisfies all constraints (Eqs. 1b-1d). ILP aims to find a feasible solution that minimizes $obj = \mathbf{c}^\top \mathbf{x}$ (Eq. 1a) in the case of a minimization problem, which defines the optimal solution. Additional properties are provided in Appendix E.1.

### 2.2 TRANSFORMER FOR GRAPHS

Existing E2EPH methods commonly utilize GCN, which is based on the message passing neural network (MPNN) (Gilmer et al., 2017). MPNNs aggregate messages from the neighbor nodes, making

them well-suited for capturing local structural information. However, they struggle to capture long-range dependencies between distant nodes (Zhang et al., 2020; Wu et al., 2021b). To propagate messages between nodes that are $K$ hops apart, an MPNN requires at least $K$ layers. In the context of ILP, capturing relationships among variables that influence each other across multiple constraints may require deep MPNNs. However, deeper architecture often suffers from the oversmoothing problem (Li et al., 2018; Wu et al., 2021b; Min et al., 2022). For instance, as shown in Figure 2(a), variables $x_2$ and $x_3$ are four hops apart: $x_2$ - $\mathbf{a_1}$ - $x_1$ - $\mathbf{a_2}$ - $x_3$. Although $x_2$ and $x_3$ do not appear in the same constraint, they are connected via $x_1$, which is shared by both $\mathbf{a_1}$ and $\mathbf{a_2}$. A change in $x_2$ can affect $x_1$, which in turn may influence $x_3$. Modeling such interactions would require four layers, but even shallow MPNNs with 2–4 layers are prone to oversmoothing (Wu et al., 2023). Thus, we design a new GNN inspired by Transformer-based GNNs (Rong et al., 2020; Ying et al., 2021; Lin et al., 2022; Min et al., 2022), which can effectively learn relationships between distant nodes (Wu et al., 2021b).

## 2.3 ACTOR–CRITIC REINFORCEMENT LEARNING

The Actor–Critic (AC) framework (Mnih et al., 2016) is a foundational RL paradigm that consists of two components: an actor, which selects actions, and a critic, which evaluates them. In this framework, the actor aims to maximize the expected reward by learning a policy $\pi_\theta(\mathcal{A} \mid \mathcal{S})$ that maps an observation $\mathcal{S}$ to a probability distribution over actions $\mathcal{A}$, while the critic evaluates $\mathcal{S}$ using a value function $V_\theta(\mathcal{S})$. Given the demonstrated effectiveness of AC for CO problems,(Bello et al., 2016; Hubbs et al., 2020; Zong et al., 2022), we adopt this to train our proposed GNN architecture.

## 2.4 NEIGHBORHOOD SEARCH FOR CO

Local search (LS) heuristic is a widely used for exploring the neighborhood of a current solution for CO problems (Hillier & Lieberman, 2015). Although LS may encounter local optima, empirical studies have shown it to be effective in quickly identifying high-quality solutions (Bertsimas & Tsitsiklis, 1997). Classic LS typically perturbs a single variable at a time, whereas exploring larger neighborhoods has been shown to improve effectiveness (Shaw, 1998). In contrast, with our newly designed search strategy, RL-SPH allows multiple variables to change simultaneously as shown in Figure 2, thereby exploring a broader neighborhood. Our search strategy resembles RENS in that it solves subproblems by fixing a set of variables, and also LNS in exploring large neighborhoods. However, unlike RENS and RL-SPH, LNS requires an *initial feasible solution*, limiting its use as a start primal heuristic (Berthold, 2006).

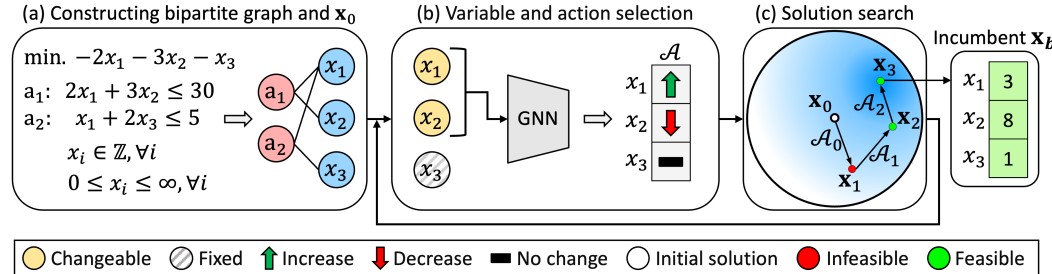

Figure 2: The overview of RL-SPH.

## 3 METHODOLOGY

This section presents RL-SPH in detail. Figure 2 illustrates the overall process. Given an ILP instance, RL-SPH constructs a bipartite graph and an initial solution $\mathbf{x}_0$. At each timestep, it selects $\tilde{n} (= 2 \lceil \log_2 n \rceil)$ changeable variables as input to the trained agent. The agent selects actions expected to yield high rewards and generates a new solution. As the process repeats, the best feasible solution found so far (i.e., the incumbent) $\mathbf{x}_b$ is updated whenever $\mathbf{x}_{t+1}$ is both feasible and improves upon $\mathbf{x}_b$.

## 3.1 REINFORCEMENT LEARNING FOR ILP

Our RL framework for ILP aims to train an agent to make decisions that maximize rewards while interacting with a given instance. Figure 3 depicts how the RL agent interacts with an ILP instance, where $\mathcal{S}_t$, $\mathcal{A}_t$, and $\mathcal{R}_{t,total}$ denote the observation, the set of selected actions, and the total reward at timestep $t$, respectively. The instance $M$ serves as the environment for the agent. Using $\mathcal{A}_t =$

$(a_{t,1}, \ldots, a_{t,n})$, the agent updates the solution $\mathbf{x}_{t+1}$ for $n$ variables. This update affects the left-hand side of the constraints $\mathbf{lhs}_{t+1}$, the feasibility state vector $\mathbf{f}_{t+1}$, and the objective value $obj_{t+1}$. At the next timestep, the agent receives a new observation $\mathcal{S}_{t+1}$ changed by its previous actions and selects a new action set $\mathcal{A}_{t+1}$ to maximize rewards. By comparing the estimated reward with the actual reward $\mathcal{R}_{t+1,total}$ obtained from $\mathcal{A}_{t+1}$, the agent refines its policy $\pi$.

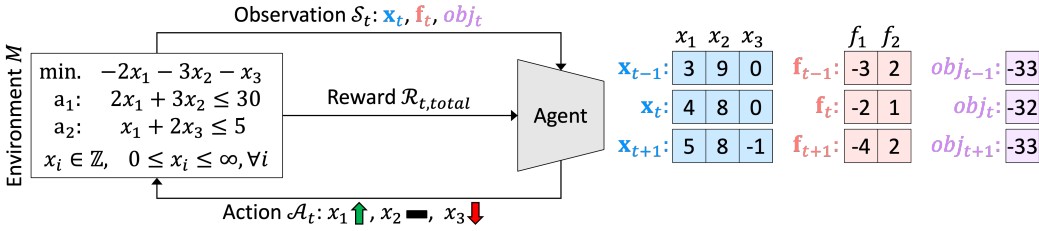

(a) Diagram of RL framework for ILP.  (b) Examples of observation for Figure 3(a).

Figure 3: Reinforcement learning for ILP.

### 3.1.1 ACTION

At timestep $t$, the agent selects a set of actions $\mathcal{A}_t = (a_{t,1}, \ldots, a_{t,n})$ for $n$ variables based on $\mathcal{S}_t$. For each variable, the agent can take one of three actions: increase, no change, or decrease, as depicted in Figure 3(a). The magnitude of change for both increases and decreases is set to 1, as discussed in Section 5. Non-changeable variables are treated as fixed (i.e., no change), as shown in Figure 2(b).

### 3.1.2 OBSERVATION

We define the observation $\mathcal{S}_t = (\mathbf{x}_t, \mathbf{f}_t, obj_t)$. The solution $\mathbf{x}_t$ is obtained by updating variable values based on the agent's previous actions $\mathcal{A}_{t-1}$. For example, if $\mathcal{A}_{t-1} = (a_{t-1,1}, a_{t-1,2}, a_{t-1,3}) = (+1, -1, +0)$, then $\mathbf{x}_{t-1}$ in Figure 3(b) is updated to $\mathbf{x}_t = (x_{t,1}, x_{t,2}, x_{t,3}) = (4, 8, 0)$. Using the updated $\mathbf{x}_t$, the new $\mathbf{lhs}_t = \mathbf{A}\mathbf{x}_t$ and $obj_t = \mathbf{c}^\top \mathbf{x}_t$ are calculated.

Each element of $\mathbf{f}_t (= \mathbf{b} - \mathbf{lhs}_t)$ indicates whether the corresponding constraint is satisfied by $\mathbf{x}_t$. Non-negative elements in $\mathbf{f}_t$ indicate satisfied constraints, while negative ones indicate violations. For example, in Figure 3, $\mathbf{x}_{t+1}$ yields $\mathbf{lhs}_{t+1} = (34, 3)$ for constraints $\mathbf{a}_1$ and $\mathbf{a}_2$. Since $\mathbf{f}_{t+1} = \mathbf{b} - \mathbf{lhs}_{t+1} = (30, 5) - (34, 3) = (-4, 2)$, $\mathbf{x}_{t+1}$ violates $\mathbf{a}_1$ but satisfies $\mathbf{a}_2$.

### 3.1.3 REWARD

We design reward functions to guide the agent in selecting actions that maximize the total reward $\mathcal{R}_{t,\text{total}}$ for a given ILP instance, as follows:

$$\mathcal{R}_{t,\text{total}} = \mathcal{R}_{t,\text{opt}} + \mathcal{R}_{t,\text{explore}} \tag{2}$$

$$\mathcal{R}_{t,\text{opt}} = \begin{cases} \mathcal{R}_{t,\text{p1}}, & \text{if agent is in } phase1, \\ \mathcal{R}_{t,\text{p2}}, & \text{otherwise.} \end{cases} \tag{3}$$

The reward system operates in two phases: $phase1$ continues until the first feasible solution is found, while $phase2$ begins thereafter. The reward functions for $phase1$ and $phase2$ are defined as follows:

$$\mathcal{R}_{t,\text{p1}} = \begin{cases} \mathcal{R}_{t,\text{bound}}, & \text{if } (x_{t+1,i} \notin [l_i, u_i] \, \exists i) \wedge \mathcal{R}_{t,\text{const}} \geq 0 \wedge obj_{t+1} < obj_t, \\ \mathcal{R}_{t,\text{bound}} - \Delta obj_t, & \text{if } (x_{t+1,i} \notin [l_i, u_i] \, \exists i) \wedge \mathcal{R}_{t,\text{const}} \geq 0 \wedge obj_{t+1} \geq obj_t, \\ \mathcal{R}_{t,\text{F}} + \Delta obj_t, & \text{if } (x_{t+1,i} \in [l_i, u_i] \, \forall i) \wedge \mathcal{R}_{t,\text{const}} \geq 0 \wedge obj_{t+1} < obj_t, \\ \mathcal{R}_{t,\text{F}} - \Delta obj_t, & \text{if } \mathcal{R}_{t,\text{const}} < 0 \wedge obj_{t+1} \geq obj_t, \\ \mathcal{R}_{t,\text{F}}, & \text{otherwise.} \end{cases} \tag{4}$$

$$\mathcal{R}_{t,\text{p2}} = \begin{cases} \Delta obj_t, & \text{if } \mathbf{x}_{t+1} \in \mathcal{F} \wedge obj_{t+1} < obj_b, \\ -\Delta obj_t \cdot \alpha, & \text{if } \mathbf{x}_{t+1} \in \mathcal{F} \wedge obj_{t+1} \geq obj_b, \\ \mathcal{R}_{t,\text{F}}, & \text{if } \mathbf{x}_{t+1} \notin \mathcal{F} \wedge obj_{t+1} < obj_b, \\ \mathcal{R}_{t,\text{F}} \cdot \alpha & \text{otherwise.} \end{cases} \tag{5}$$

$$\mathcal{R}_{t,\text{F}} = \mathcal{R}_{t,\text{bound}} + \frac{1}{\sqrt{\tilde{n}}} \mathcal{R}_{t,\text{const}} \tag{6}$$

$$\mathcal{R}_{t,\text{bound}} = -\sum_{i=1}^{n} \mathbb{I}\left(x_{t+1,i} \notin [l_i, u_i]\right), \quad \mathcal{R}_{t,\text{const}} = \sum_{j=1}^{m} \min(f_{t+1,j}, 0) - \min(f_{t,j}, 0). \quad (7)$$

where $\mathcal{F}$ is the feasible region, $\Delta obj_t = |obj_{t+1} - obj_t| / \max(|\mathbf{c}|)$, $\alpha$ is a toward-optimal bias, $\mathbb{I}$ is the indicator function, $x_{t,i}$ of $\mathbf{x}_t$ is the value of the $i$-th decision variable at timestep $t$, $f_{t,j}$ is the element of $\mathbf{f}_t$ for the $j$-th linear constraint, and $\tilde{n}$ is the number of changeable variables.

Finding a feasible solution is a prerequisite for improving the incumbent. Thus, our primary goal is to find a feasible solution that satisfies all constraints. The feasibility reward $\mathcal{R}_{t,\text{F}}$, used in Eqs. 4 and 5, is computed based on the variable bounds and linear constraints. The bound reward $\mathcal{R}_{t,\text{bound}}$ imposes a penalty proportional to the number of variables that violate their bounds. For example, in Figure 3(b), $\mathcal{R}_{t,\text{bound}} = -1$ since $l_i = 0$ and $x_{t+1,3} = -1$. The constraint reward $\mathcal{R}_{t,\text{const}}$ reflects the improvement (or deterioration) for each infeasible linear constraint. For example, in Figure 3(b), $\mathcal{R}_{t,\text{const}}$ is $[\{-4 - (-2)\} + \{0 - (0)\}] = -2$.

In $phase1$, the primary goal is to find the first feasible solution from an infeasible initial solution. Since improvements in constraint violations or objective values are meaningless when variable bounds are violated, satisfying the variable bounds should be prioritized above all else. Thus, positive values of $\mathcal{R}_{t,\text{const}}$ are ignored to prioritize satisfying the bounds (Cases 1, 2 in Eq. 4). In all other cases, the agent receives the feasibility reward $\mathcal{R}_{t,\text{F}}$ during $phase1$ (Cases 3, 4, 5 in Eq. 4). Achieving a better (i.e., lower) objective value in $phase1$ leads to a stronger starting point for $phase2$. Accordingly, the agent receives a reward proportional to the changes in the objective value (Cases 2, 3, 4 in Eq. 4). However, a positive $\Delta obj_t$ is given only when feasibility improves (Case 3 in Eq. 4), thereby preserving the priority. A well-trained agent selects actions that maximize the rewards and is thus guaranteed to discover a feasible solution, as established in Proposition 1:

**Proposition 1.** *Suppose* $\mathbf{x}_t \notin \mathcal{F}$, $\mathcal{R}_{t,const} > 0$, *and* $\mathcal{R}_{t,bound} = 0$ *for all* $t < \mathcal{T}$. *Then* $\mathbf{x}_{\mathcal{T}} \in \mathcal{F}$.

*Proof.* Appendix A provides the proof of Proposition 1. $\qquad\square$

In $phase2$, the goal is to improve the incumbent solution $\mathbf{x}_b$. If the agent finds $\mathbf{x}_{t+1} \in \mathcal{F}$, it receives a reward proportional to the changes in the objective value (Cases 1, 2 in Eq. 5). If $\mathbf{x}_{t+1} \notin \mathcal{F}$, it is penalized for violations (Cases 3, 4 in Eq. 5). A suitable $\alpha$ promotes the agent to explore promising regions where $obj_{t+1} < obj_b$. Empirically, search performance tends to increase if $\alpha > 1$; thus, we set $\alpha$ to 2. Appendix J.5.2 presents an evaluation of the impact of toward-optimal bias.

Exploration must take precedence even over satisfying variable bounds, as the agent might otherwise remain stationary to avoid penalties. To discourage this behavior, we assign a heavy penalty of $-100$ for the exploration penalty $\mathcal{R}_{t,\text{explore}}$ in Eq. 2 only if $\mathbf{x}_{t+1} = \mathbf{x}_t$. The maximum bound penalty occurs when all changeable variables violate their bounds, expressed as $-\tilde{n} = -2\lceil \log_2 n \rceil$. Exceeding $-100$ would require a bound penalty corresponding to a large $n > 10^{15}$, which is practically infeasible. Thus, we impose $-100$ if the agent does not move. Appendix H provides additional explanation on the reward functions, including the underlying motivation and illustrative examples.

## 3.2 GNN ARCHITECTURE OF THE RL AGENT

Figure 4 shows our GNN based on a Transformer encoder. The actor layer generates a sequence of actions conditioned on the selected $\tilde{n}$ variables and their structural information $(\mathbf{c}^\top | \mathbf{A})$. To stabilize training, we scale $(\mathbf{c}^\top | \mathbf{A})$ to $[-1, 1]$ using equilibration scaling (Tomlin, 1975), which normalizes each constraint by its largest absolute coefficient while preserving problem equivalence. The scaled structural information is projected to the hidden size. A binary feature $bnd\_lim$ is set to 1 if a variable value reaches or exceeds its bound, otherwise 0. For example in Figure 3, the $bnd\_lim$ for $x_{t,1}$, $x_{t,2}$, and $x_{t,3}$ would be 0, 0, and 1, respectively. Since the raw variable values

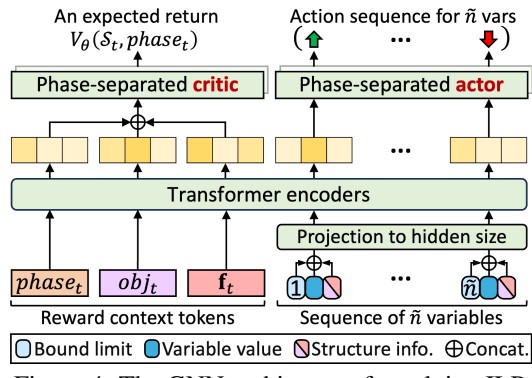

Figure 4: The GNN architecture for solving ILP.

may be unbounded, we embed them using Periodic Embedding (PE) (Gorishniy et al., 2022), which has proven effective for numerical features in ML tasks (Pace & Barry, 1997; Kohavi et al., 1996). PE

is formulated as $\mathrm{PE}(z) = \oplus(\sin(\tilde{z}), \cos(\tilde{z}))$, where $\tilde{z} = [2\pi w_1 z, \ldots, 2\pi w_k z]$, $\oplus(\cdot)$ is concatenation, with scalar $z$ and trainable $w_i$. Each final variable node is obtained by concatenating its structural information, $bnd\_lim$, and variable value embedding, followed by a projection to the hidden size.

The critic layer approximates an expected return using reward context tokens, which consist of a phase indicator $phase_t$, a PE-encoded $obj_t$, and a scaled $\mathbf{f}_t$. The reward context is essential since the total reward is calculated based on improvements in solution quality and feasibility depending on the phase (see Eq. 2). Since the two phases have distinct goals, $phase_t$ informs the agent of its current phase. We introduce *phase-separated actor* and *critic layers* to align with the two-phase reward designs. In both phases, all other layers share parameters, and the input/output configuration is consistent. $\mathbf{f}_t$ is scaled by $\sqrt{|\mathbf{b}| + |\mathbf{b} - \mathbf{lhs}_t|}$ and projected to the hidden size. The reward context tokens and variable nodes constitute a sequence of length $\tilde{n} + 3$, which is fed into our Transformer.

## 3.3 FEASIBILITY-AWARE SEARCH STRATEGY

To explore the solution space efficiently, RL-SPH selects $\tilde{n} = p + q$ changeable variables that are more likely to improve feasibility at each step $t$. The selection process comprises two steps, varying by phase. In $phase1$, RL-SPH stochastically selects $p$ seed variables that frequently appear in violated linear constraints. Then, it selects the top $q$ neighboring variables that appear most often with the seed variables in the same constraints. For example, in Figure 3, at $t$, constraint $\mathbf{a}_1$ is infeasible, so $x_1$ and $x_2$ are candidates for selection. If $p = 1$, $q = 1$, and $x_2$ is selected as the seed, then $x_1$, which appears in the same constraint, is selected as a neighbor. Thus, $x_1$ and $x_2$ become changeable, while $x_3$ remains fixed. In $phase2$, RL-SPH stochastically selects $p$ seed variables that frequently appear in constraints with low violation risk. For instance, if $\mathbf{f}_t = (0, 5)$, then $\mathbf{a}_2$ has a lower violation risk. Neighbor selection in $phase2$ follows the same procedure as in $phase1$. To limit the growth of input size in the GNN, we set $p$ and $q$ to $\lceil \log_2 n \rceil$. The pseudo-code appears in Algorithm 3, Appendix D.

Algorithm 1 outlines the overall procedure of our search strategy, which is used during both training and testing. RL-SPH first selects the variables to be changed with the feasibility-aware selection (Line 1), which are then used by the actor layer $\pi_\theta$ to predict $\mathcal{A}_t$ (Line 2). Based on $\mathcal{A}_t$, the agent obtains the next observation $\mathcal{S}_{t+1}$ (Lines 3-4). The agent receives a reward determined by the quality of $\mathbf{x}_{t+1}$ and the degree of feasibility improvement (Line 5). If $\mathbf{x}_{t+1}$ is a better feasible solution, both $obj_b$ and $\mathbf{x}_b$ are updated (Lines 6-8). To guide exploration, we restrict movement to areas where further search is unnecessary (e.g., bound violations) (Lines 9–11). In $phase1$, the agent explores freely the solution space unless variable bounds are violated (Lines 9–10). In $phase2$, movements are rolled back unless a better feasible solution is discovered (Lines 9–10). The algorithm returns $\mathcal{R}_{t,\text{total}}$, $\mathcal{S}_{t+1}$, $\mathbf{x}_b$, and $obj_b$ for the next step (Line 12). Appendix K.5 provides a discussion of our search strategy.

---

**Algorithm 1** Feasibility-aware solution search of RL-SPH

**Input:** instance $M$, actor layer $\pi_\theta$, current observation $\mathcal{S}_t = (\mathbf{x}_t, \mathbf{f}_t, obj_t)$, incumbent solution $\mathbf{x}_b$ and value $obj_b$, current phase $\texttt{phase}_t$
**Output:** reward $\mathcal{R}_{t,\text{total}}$, new observation $\mathcal{S}_{t+1}$, incumbent solution $\mathbf{x}_b$, and value $obj_b$

1: $\tilde{\mathcal{S}}_t \leftarrow \text{select\_variable}(M, \mathcal{S}_t, \texttt{phase}_t)$          {See Algorithm 3}
2: $\mathcal{A}_t \leftarrow \pi_\theta(\tilde{\mathcal{S}}_t, \texttt{phase}_t)$          {See Section 3.2}
3: $\mathbf{x}_{t+1} \leftarrow \text{move}(\mathbf{x}_t, \mathcal{A}_t)$          {See Section 3.1.1}
4: $\mathcal{S}_{t+1} \leftarrow \text{observe}(M, \mathbf{x}_{t+1})$          {See Section 3.1.2}
5: $\mathcal{R}_{t,\text{total}} \leftarrow \text{reward}(M, \mathcal{S}_{t+1}, \mathcal{S}_t, obj_b, \texttt{phase}_t)$          {See Equation 2}
6: **if** $\mathbf{x}_{t+1} \in \mathcal{F}$ **and** $obj_{t+1} < obj_b$ **then**
7:      $obj_b \leftarrow obj_{t+1}$
8:      $\mathbf{x}_b \leftarrow \mathbf{x}_{t+1}$
9: **else if** ($\texttt{phase}_t = 1$ **and** $x_i \notin [l_i, u_i], \exists i$) **or** $\texttt{phase}_t = 2$ **then**
10:      $\mathcal{S}_{t+1} \leftarrow \mathcal{S}_t$
11: **end if**
12: **return** $\mathcal{R}_{t,\text{total}}, \mathcal{S}_{t+1}, \mathbf{x}_b, obj_b$

---

**Proposition 2.** *If $\mathcal{T} \leq n$, then RL-SPH finds a feasible solution for ILP within polynomial time.*

*Proof.* Appendix B provides the proof of Proposition 2.      □

## 3.4 LEARNING ALGORITHM

During training, our agent begins with an initial solution obtained via either *LP-relaxation* or *random assignment*. If a more sophisticated initialization method is available, it can be readily integrated into

RL-SPH. At timestep $t$, the agent observes $\tilde{\mathcal{S}}_t$ with $\tilde{n}$ changeable variables. Afterwards, it selects $\mathcal{A}_t$ using $\pi_\theta(\mathcal{A}_t \mid \tilde{\mathcal{S}}_t, phase_t)$. Upon executing $\mathcal{A}_t$, the ILP environment returns the observed reward $\mathcal{R}_{t,\text{total}}$ and $\mathcal{S}_{t+1}$. The actor is trained to encourage actions that yield $\mathcal{R}_{t,\text{total}} > V_\theta(\mathcal{S}_t, phase_t)$ and to discourage those that yield lower rewards. The critic is trained to minimize the gap between $\mathcal{R}_t$ and $V_\theta(\mathcal{S}_t, phase_t)$ to provide accurate feedback to the actor. The agent stays in $phase1$ for a predefined number of steps to ensure sufficient training, even after finding the first feasible solution. Once the step limit is reached, it moves to a new instance. Appendix C details the full training procedure.

## 4 EXPERIMENTS

In this section, we validate the effectiveness of RL-SPH through four experiments. First, we compare RL-SPH with existing start primal heuristics to demonstrate its ability to quickly find high-quality feasible solutions. Second, we evaluate RL-SPH combined with an ILP solver against SCIP and the existing E2EPH (Han et al., 2023), to examine whether it can reach high-quality solutions *more quickly*. Third, we assess the generalizability of RL-SPH in cross-problem settings. Fourth, we conduct ablation studies to evaluate the effectiveness of our feasibility-aware search strategy and our GNN's components. Appendix J presents additional experiments, including comparisons with SOTA E2EPH, further generalization tests, hyperparameter analyses, qualitative analyses, and more.

### 4.1 EXPERIMENTAL SETUP

#### 4.1.1 BENCHMARKS

We conduct experiments on five NP-hard ILP benchmarks commonly used in prior works (Gasse et al., 2019; Han et al., 2023; Huang et al., 2023; Qi et al., 2021). For minimum vertex cover (MVC), we generate instances based on the Barabási-Albert random graph models (Albert & Barabási, 2002), with 3,000 nodes, yielding 3,000 variables and 11,931 constraints on average. We generate instances for independent set (IS) using the same model as MVC with 1,500 nodes. For set covering (SC) (Balas & Ho, 1980), we generate instances with 3,000 variables and 2,000 constraints. For combinatorial auction (CA) (Leyton-Brown et al., 2000), instances are generated with 4,000 items and 2,000 bids, resulting in 4,000 variables and 2,677 constraints on average. We generate instances with 2,000 integer variables and 2,000 constraints following (Qi et al., 2021), denoted as non-binary integers (NBI). For each dataset, we generated 1,000 training instances and 100 test instances. Additional details on the datasets are given in Appendix I.1.

#### 4.1.2 BASELINES

In the first experiment, we compare RL-SPH against four commonly used start primal heuristics, as detailed in Appendix E.2: Diving, FP, Rounding, and RENS. We use the built-in implementations from the open-source ILP solver SCIP (v8.1.0), with presolving and branching disabled to isolate the heuristic performance. SCIP's 15 diving heuristics are grouped under Diving, and its six rounding heuristics under Rounding. For each baseline, only the corresponding heuristic is enabled with default setting, while all other heuristics are disabled. Each baseline runs until its own termination condition is met, with a maximum time limit of 1,000 seconds. RL-SPH, which has no predefined termination condition, stops when all baseline methods complete their search. If RL-SPH is terminated without any feasible solution, it would be treated as a failure. In the second experiment, we evaluate RL-SPH combined with SCIP (**RL-SPH+S**) against two baselines: PAS (Han et al., 2023), a representative E2EPH, combined with SCIP (**PAS+S**) and SCIP. Both SCIP and PAS+S are run with their default settings. PAS+S is trained the same duration as RL-SPH, taking around 31 minutes on IS as shown in Appendix I.3. All randomization parameters in SCIP are set to 0 for reproducibility. Unless otherwise stated, RL-SPH uses LP initialization, and its termination criteria follow those of the first experiment.

#### 4.1.3 METRICS

We use four evaluation metrics for the main experiment: the *primal gap* (PG) (Huang et al., 2023; Cantürk et al., 2024), the *primal integral* (PI) (Gasse et al., 2022), the *feasibility rate* (FR), and the *first feasible solution time* (FT). PG quantifies how close a method's incumbent value $obj_b$ is to the best-known solution ($BKS$), and is computed as $\text{PG}(obj_b) = \frac{|obj_b - BKS|}{\max(|obj_b|, |BKS|, \epsilon)} * 100$. PI measures how quickly the incumbent improves toward $BKS$ over time, and is defined as $\text{PI}(T) = \sum_{t=1}^{T} \text{PG}(obj_t)$, where $\text{PG}(obj_t) = 1$ if no feasible solution is available at actual wall-clock time $t$ (in seconds). Both PG and PI are averaged only over instances where at least one feasible solution is found. FR measures the ratio of instances in which a feasible solution is obtained. FT records the time taken to obtain the first feasible solution, measured in seconds.

## 4.2 COMPARISON WITH START PRIMAL HEURISTICS

Table 1 presents the evaluation results in terms of FR, PG, PI, and FT, where $BKS$ is the best objective value among all methods. RL-SPH achieved 100% FR on all benchmarks, demonstrating its effectiveness in learning feasibility. Among the baselines, only Rounding also achieved 100% FR on all benchmarks. We denote RL-SPH initialized with LP as RL-SPH(LP), and RL-SPH initialized with a randomly generated solution as RL-SPH(Random). Compared to the PG values of the baselines with 100% FR, RL-SPH(LP) achieved a 44× lower PG on average, indicating its superiority in discovering high-quality feasible solutions. It also attained a 2.3× lower PI on average, suggesting faster converges toward such solutions. Notably, RL-SPH(Random) showed comparable performance to RL-SPH(LP), which begins from an LP-feasible solution. Furthermore, RL-SPH(Random) found the first feasible solution within 2 seconds across all datasets. Random initialization enables an early start, accelerating the agent's exploration for feasible solutions without waiting for LP solving. These results suggest that RL-SPH can find feasible solutions even from less accurate initializations, and motivate the development of an initialization strategy that is more refined than random assignment but faster than LP-relaxation. RENS and Diving yielded similar results on IS and MVC, as both start from the same LP-feasible solution due to the same random seed for reproducibility. Moreover, the coefficient matrix $\mathbf{A}$ in these datasets is extremely sparse (density of 0.13% and 0.07%), resulting in LP-feasible solutions with few fractional values. In such cases, fixing variables has limited impact on the rest of the problem, leading both methods to produce similar solutions. Figure 5 shows that RL-SPH reaches the best objective value, outperforming the other methods with a 100% FR. This demonstrates that RL-SPH achieves its goal of quickly obtaining high-quality feasible solutions. In Appendix J.1, we also compare RL-SPH with the most recent E2EPH.

Table 1: Performance comparison among the start primal heuristics on four benchmarks. **Bold** and underline are used to indicate the best and second-best methods among those with 100% FR.

| Dataset | Metric | Diving | FP | Rounding | RENS | RL-SPH (Ours) | |
|---|---|---|---|---|---|---|---|
| | | | | | | w/ Random | w/ LP |
| IS | FR (%) ↑ | 89 | 100 | 100 | 89 | 100 | 100 |
| | PG (%) ↓ | 99.98±0.05 | 20.60±2.10 | 18.09±2.74 | 99.98±0.05 | 4.10±2.25 | **0.14±0.59** |
| | PI ↓ | 19.1±2.3 | 4.8±0.7 | 4.9±0.7 | 19.1±2.3 | 3.8±0.4 | **2.5±0.2** |
| | FT (sec) ↓ | 11.94±2.27 | 0.22±0.04 | **0.10±0.00** | 11.94±2.25 | 0.27±0.07 | 0.55±0.07 |
| CA | FR (%) ↑ | 100 | 100 | 100 | 100 | 100 | 100 |
| | PG (%) ↓ | 90.61±1.68 | 18.09±9.64 | 12.02±9.97 | 81.70±30.19 | 19.04±8.20 | **3.82±9.77** |
| | PI ↓ | 105.7±27.9 | 31.5±15.6 | 23.0±15.5 | 112.9±27.7 | 36.7±14.9 | **21.9±13.8** |
| | FT (sec) ↓ | 10.06±0.90 | 10.66±0.70 | 8.03±0.51 | 102.17±28.38 | **0.41±0.08** | 8.26±0.72 |
| SC | FR (%) ↑ | 1 | 99 | 100 | 3 | 100 | 100 |
| | PG (%) ↓ | 0.00±0.00 | 0.34±3.11 | 29.98±12.38 | 0.00±0.00 | 12.78±16.25 | **9.67±15.41** |
| | PI ↓ | 60.0±0.0 | 347.5±73.5 | 321.5±111.0 | 339.3±212.8 | **101.3±136.7** | 168.0±124.9 |
| | FT (sec) ↓ | 59.50±0.00 | 344.75±68.57 | 104.09±16.97 | 338.50±212.62 | **0.19±0.07** | 95.43±7.39 |
| MVC | FR (%) ↑ | 36 | 100 | 100 | 36 | 100 | 100 |
| | PG (%) ↓ | 33.67±0.70 | 6.76±1.15 | 8.01±1.20 | 33.67±0.70 | **0.23±0.45** | 0.81±0.86 |
| | PI ↓ | 46.0±5.3 | 5.2±1.2 | 6.1±1.2 | 42.8±5.2 | **4.2±0.5** | 5.0±0.6 |
| | FT (sec) ↓ | 40.37±7.05 | 1.17±0.24 | **0.20±0.04** | 35.76±6.95 | 1.63±0.07 | 1.55±0.08 |

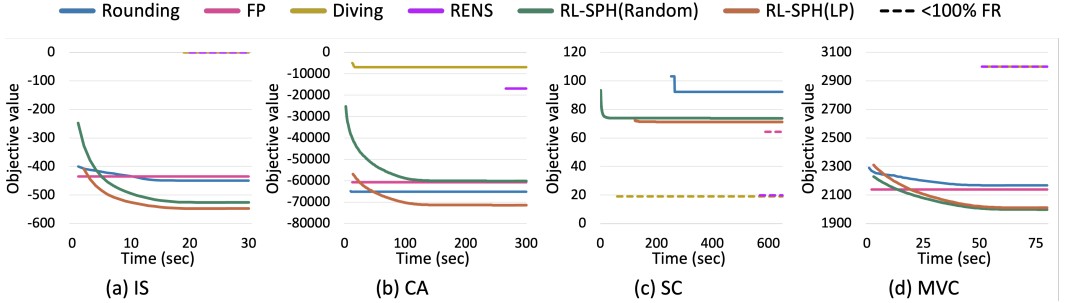

Figure 5: Illustration of the objective values of the compared methods over time (in seconds). Lines indicate valid averaged incumbent objective values, which are computed only when an incumbent objective value exists for all instances at a given time point. Dashed lines indicate that the FR is not 100%, and the averaging includes only instances for which at least one feasible solution was found.

### 4.3 EVALUATION OF RL-SPH COMBINED WITH AN ILP SOLVER

Table 2 presents the optimization performance of the compared methods under a 50-second time limit, where $BKS$ was obtained by SCIP within 1,000 seconds. RL-SPH+S fixes each variable to a unanimous value based on the feasible solutions generated by RL-SPH(LP) during the first five seconds. RL-SPH+S outperformed both SCIP and PAS+S in terms of PG and PI on both benchmarks, achieving high-quality solutions with a PG below 1% relative to the $BKS$.

Table 2: Solving performance of RL-SPH combined with SCIP.

| Dataset | Metric | SCIP | PAS+S | **RL-SPH+S** |
|---------|--------|------|-------|--------------|
| IS | FR (%) ↑ | 100 | 100 | **100** |
| | PG (%) ↓ | 1.24±1.01 | 0.30±0.61 | **0.14±0.13** |
| | PI ↓ | 2.4±0.3 | 1.7±0.4 | **0.5±0.5** |
| NBI | FR (%) ↑ | 100 | *not applicable* | **100** |
| | PG (%) ↓ | 0.24±0.03 | | **0.23±0.04** |
| | PI ↓ | 11.5±2.3 | | **4.9±0.9** |

Unlike PAS, RL-SPH is capable of generating feasible solutions independently, allowing it to safely fix more variables using an ensemble of feasible solutions, which results in a more reduced problem size. Moreover, RL-SPH supports non-binary integer variables with a 2.3× lower PI than SCIP, whereas PAS does not support such variables (NBI in Table 2). **Limitation**: Although reducing the problem size can significantly accelerate the discovery of high-quality solutions (Nair et al., 2020), it may also introduce sub-optimality. Given that the primary goal of primal heuristics is to quickly find high-quality feasible solutions (Berthold, 2006; Cantürk et al., 2024), exact optimality is often considered a secondary concern. Nevertheless, improving the optimality within RL-SPH could be a promising direction for future research, as it remains an open challenge of ML for CO (Son et al., 2023). We discuss directions for improving RL-SPH's optimality in Appendix K.10.

### 4.4 EVALUATION OF CROSS-PROBLEM GENERALIZATION

Table 3 presents the performance of RL-SPH in cross-problem settings. In this experiment, we matched the number of variables in MVC to that of IS ($n$=1,500) to control for factors other than the problem type. RL-SPH trained solely on IS failed to produce any feasible solution on MVC test instances and vice versa. In contrast, a joint RL-SPH model trained on both problem types achieved a 100% FR on each dataset. Moreover, the joint model achieved better incumbent values than Rounding, which is the strongest baseline in terms of FR (see Appendix K.1).

Table 3: Cross-problem performance of RL-SPH. *fail* indicates that the metrics cannot be computed because no feasible solution was obtained (i.e., FR = 0%).

| Test set | Metric | Rounding | RL-SPH trained on | | |
|----------|--------|----------|------|------|--------|
| | | | IS | MVC | IS+MVC |
| IS | FR (%) ↑ | 100 | 100 | 0 | 100 |
| | PG (%) ↓ | 17.65±2.87 | 0.00±0.00 | *fail* | 15.55±2.82 |
| | PI ↓ | 4.6±0.7 | 2.4±0.2 | | 5.1±0.6 |
| MVC | FR (%) ↑ | 100 | 0 | 100 | 100 |
| | PG (%) ↓ | 9.06±1.58 | *fail* | 0.00±0.00 | 7.13±1.56 |
| | PI ↓ | 2.8±0.4 | | 1.8±0.2 | 2.8±0.4 |

We further evaluated the joint RL-SPH model on several real-world instances from the MIPLIB benchmark (Gleixner et al., 2021). Table 4 presents summary statistics of the instances used in this experiment along with incumbent values obtained by RL-SPH and Rounding. RL-SPH obtains feasible solutions for all instances, even though the instances differ entirely from the training instances for the joint model in terms of density, constraint type, and size. In contrast, Rounding, the strongest baseline in terms of FR, fails to find a feasible solution for `neos-3530903-gauja`. Notably, `neos-3530903-gauja` contains a mixture of binary and integer variables, which differs substantially from the training instances (IS and MVC) that include only binary variables. These results demonstrate that RL-SPH can generalize problem-solving capabilities to unseen instances even when the variable types differ. They further highlight RL-SPH's potential to function as a unified solver across diverse problem settings, which is highly desirable in real-world CO (Liu et al., 2024a).

Table 4: Statistics of instances from the official MIPLIB website and the incumbent values of RL-SPH(IS+MVC) and Rounding. To the best of our knowledge, the website does not provide information on how the $BKS$ was obtained and how much computation time was used. The time limit for Rounding and RL-SPH is set to 100 seconds.

| Test instance | Type | #binaries | #integers | #constraints | Density | $BKS$ ↓ | Rounding | RL-SPH |
|---------------|------|-----------|-----------|--------------|---------|---------|----------|--------|
| cod105 | set packing | 1,024 | 0 | 1,024 | 5.47% | -12 | -2 | -6 |
| queens-30 | knapsack | 900 | 0 | 960 | 10.81% | -40 | -27 | -24 |
| gen-ip054 | general linear | 0 | 30 | 27 | 65.7% | 6840.97 | 7148.68 | 7061.01 |
| neos-3530903-gauja | mixed binary | 420 | 1,890 | 220 | 0.87% | 168 | *infeasible* | 476 |

## 4.5 ABLATION STUDY

Table 5 presents the effectiveness of our variable selection strategy across five datasets. We introduce $PG_f$ and $\mathcal{T}$ as evaluation metrics to analyze the first feasible solution. $PG_f$ is a variant of PG in which the incumbent value $obj_b$ is replaced by the first feasible value, measuring the quality of the first feasible solution. $\mathcal{T}$ represents the number of search steps to find the first feasible solution. Across all datasets, RL-SPH consistently obtained a feasible solution with the feasibility-aware selection strategy. In contrast, with the uniform selection, it often struggled to find a feasible solution quickly and even failed within the 100-second time limit on CA. For NBI, while RL-SPH required more steps with the feasibility-aware strategy, it achieved a higher-quality first feasible solution. These results demonstrate that our search strategy is effective in quickly finding high-quality feasible solutions.

Table 5: Ablation study on our solution search strategy. Uniform denotes variable selection uniformly at random, and F-aware refers to the feasibility-aware variable selection.

| Metric | IS | | CA | | SC | | MVC | | NBI | |
|---|---|---|---|---|---|---|---|---|---|---|
| | Uniform | F-aware | Uniform | F-aware | Uniform | F-aware | Uniform | F-aware | Uniform | F-aware |
| FR (%) ↑ | 100 | 100 | 0 | 100 | 100 | 100 | 100 | 100 | 100 | 100 |
| $PG_f$ (%) ↓ | 95.3±1.6 | **41.4±1.9** | *fail* | **31.4±4.4** | 52.3±4.8 | **39.0±4.3** | 33.8±0.7 | **16.0±0.9** | 0.6±0.2 | **0.3±0.1** |
| $\mathcal{T}$ ↓ | 399±98 | **36±2** | | **59±7** | 6±1 | **4±1** | 613±109 | **68±2** | **27±6** | 46±11 |

Table 6 presents an ablation study on the GNN architecture, where $BKS$ is defined as the best objective value among all models within a 50-second time limit. All models were trained under the same configuration. $\emptyset$ showed the lowest performance on both benchmarks and notably failed to find any feasible solution on CA. With TE, the model achieved a 100% FR on both benchmarks, indicating its effectiveness in learning feasibility. Combining all components achieved the best results in PG and PI, highlighting the importance of each component. Appendix J.6.2 provides a qualitative analysis of long-range variable dependencies in TE.

Table 6: Ablation study on the proposed GNN architecture: RC (reward context), PSL (phase-separated layers), and TE (transformer encoder). $\emptyset$ denotes models without all components.

| Dataset | Metric | $\emptyset$ | {RC, PSL} | {TE, PSL} | {TE, RC} | **{TE, RC, PSL}** |
|---|---|---|---|---|---|---|
| IS | FR (%) ↑ | 100 | 100 | 100 | 100 | **100** |
| | PG (%) ↓ | 41.23±1.95 | 33.98±2.45 | 1.10±1.40 | 12.93±1.96 | **0.62±1.08** |
| | PI ↓ | 21.8±0.9 | 28.9±2.4 | 4.4±0.7 | 11.7±0.9 | **4.1±0.5** |
| CA | FR (%) ↑ | 0 | 0 | 100 | 100 | **100** |
| | PG (%) ↓ | *fail* | *fail* | 3.00±2.27 | 1.63±1.96 | **1.20±1.53** |
| | PI ↓ | | | 12.4±1.1 | 12.5±1.0 | **11.8±0.8** |

## 5 DISCUSSION ON SCALABILITY AND MOVEMENT MAGNITUDE

Recent studies have shown that breaking a large problem into smaller ones can effectively address large-scale ILP (Ye et al., 2023; 2024). This finding suggests that, even if RL-SPH may not directly handle large-scale ILP, it has the potential to solve them by decomposing the original large-scale problem into smaller ones. We also evaluated scalability more thoroughly in Appendix J.2.2.

In our preliminary experiments, we observed that increasing the movement magnitude led to better objective values. Despite this finding, we set the action space of RL-SPH to {-1, 0, +1} in our main experiments, as our focus is on feasibility rather optimality (see Appendix K.1). Further details on the magnitude are provided in Appendix K.9. We discuss various aspects of RL-SPH in Appendix K.

## 6 CONCLUSION

We proposed RL-SPH capable of independently generating high-quality feasible solutions for ILP. RL-SPH theoretically guarantees feasibility even from initially infeasible solutions. Empirically, RL-SPH achieves a 100% feasibility rate across five benchmarks, outperforming existing start primal heuristics with an average of 44× lower primal gap and 2.3× lower primal integral. Given that feasibility is a crucial prerequisite for optimality, our research on guaranteeing feasibility represents a meaningful advance toward addressing the major challenge of optimality in ML for CO.

**Appendix Summary.** The appendix includes proofs, algorithmic pseudo-code, additional preliminaries, related work, positioning relative to prior work, further explanation of the reward design, details of experimental setup, and supplementary experimental results and discussion.

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

# Appendix contents

# A PROOF OF PROPOSITION 1

In this section, we prove Proposition 1:

*Suppose $\mathbf{x}_t \notin \mathcal{F}$, $\mathcal{R}_{t,const} > 0$, and $\mathcal{R}_{t,bound} = 0$ for all $t < \mathcal{T}$. Then $\mathbf{x}_{\mathcal{T}} \in \mathcal{F}$.*

## A.1 TERM DEFINITIONS

- $\mathcal{F}$: The set of feasible solutions (feasible region) for the given original ILP problem.
- $\mathbf{x}_t$: The solution of the problem at timestep $t$.
- $\mathcal{R}_{t,\text{const}}$: The constraint reward at timestep $t$.
- $\mathcal{R}_{t,\text{bound}}$: The bound reward which is the sum of $\mathcal{R}_{t,\text{ub}}$ and $\mathcal{R}_{t,\text{lb}}$ at timestep $t$.
- $f_{t,j}$: The feasibility state for the $j$-th linear constraint at timestep $t$.
- $lhs_{t,j}$: The left-hand side of the $j$-th linear constraint at timestep $t$.
- $b_j$: The right-hand side of the $j$-th linear constraint.

## A.2 BACKGROUND

In our system, the action policy guarantees that the integrality requirements (Eq. 1c) are satisfied (see Section 3.1.1). Therefore, we can safely ignore the integrality constraints. In addition, since $phase1$ continues until the first feasible solution is found by the definition for the phases in Section 3.1.3, we focus on satisfying the linear constraints (Eq. 1b) and the bound constraints (Eq. 1d) in $phase1$.

Since a well-trained agent selects actions that maximize the reward, the rewards $\mathcal{R}_{t,\text{const}}$ and $\mathcal{R}_{t,\text{bound}}$ are expected to be positive and zero, respectively, according to the definition in Eq. 7. Therefore, we suppose that $\mathcal{R}_{t,\text{const}} > 0$ and $\mathcal{R}_{t,\text{bound}} = 0$ during $phase1$ (i.e., exploration starting from an infeasible solution).

## A.3 PROOF

Suppose $\mathbf{x}_t \notin \mathcal{F}$, $\mathcal{R}_{t,\text{const}} > 0$ and $\mathcal{R}_{t,\text{bound}} = 0$.

$$\mathcal{R}_{t,\text{const}} = \sum_{j=1}^{m} \min(f_{t+1,j}, 0) - \min(f_{t,j}, 0) > 0$$

This implies:

$$\sum_{j=1}^{m} \min(f_{t,j}, 0) < \sum_{j=1}^{m} \min(f_{t+1,j}, 0)$$

Applying the same reasoning for the next timestep $t + 1$:

$$\sum_{j=1}^{m} \min(f_{t+1,j}, 0) < \sum_{j=1}^{m} \min(f_{t+2,j}, 0)$$

Thus, the sequence $\sum_{j=1}^{m} \min(f_{t,j}, 0)$ is monotonically increasing and upper-bounded by 0, which leads to:

$$\lim_{t \to \mathcal{T}} \sum_{j=1}^{m} \min(f_{t,j}, 0) = 0$$

Considering the $\min(\cdot)$ function at timestep $\mathcal{T}$, the linear constraints (Eq. 1b) satisfied as:

$$f_{\mathcal{T},j} = b_j - lhs_{\mathcal{T},j} \geq 0, \forall j$$

$$lhs_{\mathcal{T},j} \leq b_j, \forall j$$

$$\mathbf{A}\mathbf{x}_{\mathcal{T}} \leq \mathbf{b}$$

For $\mathcal{R}_{t,\text{bound}}$, a value of zero indicates that no decision variable violates its bounds (see Eq. 7). Hence, $\mathbf{x}_{\mathcal{T}} \in \mathcal{F}$ (i.e., $\mathbf{x}_{\mathcal{T}}$ is feasible) as long as $\mathcal{R}_{t,\text{const}} > 0$ and $\mathcal{R}_{t,\text{bound}} = 0$ in $phase1$.

## B    PROOF OF PROPOSITION 2

The time complexity for RL-SPH to obtain the first feasible solution is the sum of the initial solution construction and the feasible solution exploration, yielding a total time complexity of $T_{total} = T_{init} + O\left(\mathcal{T} \cdot \{mn + \tilde{n}^2\}\right)$, where $\mathcal{T}$ is the number of search steps to find the first feasible solution (see Proposition 1). In RL-SPH, the initial solution can be obtained via either *random assignment* or *LP-relaxation*, and we base our analysis on *LP-relaxation* which incurs higher complexity. Thus, $T_{init} = T_{LP}$ in this analysis, which is solvable in polynomial time (Karmarkar, 1984). An LP-feasible solution may not be feasible for the original ILP due to the violation of the integrality constraints (see Appendix E.1). Since the case where the initial solution is ILP-feasible is trivial, we assume that the initial solution obtained from the LP-relaxation is ILP-infeasible. After initialization, the RL agent searches for a feasible solution. The agent selects $\tilde{n}$ variables ($O\left(mn\right)$ by $\tilde{\mathbf{f}}_t^\top \tilde{\mathbf{A}}$ in Algorithm 3). It then processes the selected variables via the Transformer ($O\left(\tilde{n}^2\right)$ (Katharopoulos et al., 2020)). Finally, it computes the next observation based on the selected variables ($O\left(m\tilde{n}\right)$). Thus, the complexity of Algorithm 1 is $O(mn + \tilde{n}^2)$. Suppose that $\mathcal{T} \leq n$, which is reasonable because at most $n$ variables are involved in the violations and RL-SPH can modify multiple variable values in a single search step. This condition $\mathcal{T} \leq n$ does not force the agent to improve feasibility at every step and thus allows stalls. Empirical results further support our justification for this condition, showing that $\mathcal{T}$ is much smaller than $n$ (see Table 5). Even when $\mathcal{T} = n$, the agent's search remains polynomial time. Hence, RL-SPH finds the first feasible solution for ILP in polynomial time when $\mathcal{T} \leq n$. We provide a time complexity comparison with widely used start primal heuristics in Appendix K.2.

## C    PSEUDO-CODE FOR LEARNING ALGORITHM

We obtain the initial solution (Line 2) using two methods: LP-relaxation and random assignment. For LP-relaxation, we apply random rounding to the LP-feasible solution to convert the fractional variable values into integers, which may result in an infeasible solution for the original ILP problem. For random assignment, when training on the first instance, we randomly assign the value 1 to 1% of the variables. For subsequent instances, we assign the value 1 to $r$ randomly selected variables, where $r$ is half the number of variables with non-zero values in the previous instance.

---

**Algorithm 2** Learning a policy for RL-SPH

---

**Input:** agent parameters $\theta$, instance $M$
**Parameter:** update limit $N$, total step limit $T_{max}$, step limit for $phase1$ $T_{stay}$
**Output:** updated parameters $\theta$

1: **for** $N$ updates **do**
2:     $\mathbf{x}_0 \leftarrow$ get_initial_solution$(M)$
3:     $\mathcal{S}_0 \leftarrow$ observe$(M, \mathbf{x}_0)$
4:     $\mathbf{x}_b \leftarrow \varnothing$
5:     $obj_b \leftarrow \infty$
6:     $\texttt{phase}_0 \leftarrow 1$
7:     $\texttt{stay} \leftarrow \texttt{True}$
8:     **for** $t = 0, 1, 2, \ldots, T_{max}$ **do**
9:         $\mathcal{R}_{t,\text{total}}, \mathcal{S}_{t+1}, \mathbf{x}_b, obj_b \leftarrow$ search$(M, \pi_\theta, \mathcal{S}_t, \mathbf{x}_b, obj_b, \texttt{phase}_t)$          {See Algorithm 1}
10:         **if** $\texttt{stay} = \texttt{True}$ **and** $(\mathbf{x}_{t+1} \in \mathcal{F}$ **or** $t = T_{stay})$ **then**
11:             $\mathcal{S}_{t+1} \leftarrow \mathcal{S}_0$
12:             $\mathbf{x}_b \leftarrow \varnothing$
13:             $obj_b \leftarrow \infty$
14:             $\texttt{stay} \leftarrow \texttt{False}$
15:         **else if** $\texttt{stay} = \texttt{False}$ **and** $\mathbf{x}_{t+1} \in \mathcal{F}$ **then**
16:             $\texttt{phase}_{t+1} \leftarrow 2$
17:         **end if**
18:         $\delta_{td} \leftarrow \mathcal{R}_{t,\text{total}} + \gamma \cdot V_\theta(\mathcal{S}_{t+1}, \texttt{phase}_{t+1}) - V_\theta(\mathcal{S}_t, \texttt{phase}_t)$
19:         $\mathcal{L}_\theta \leftarrow -\log \pi_\theta(\mathcal{A}_t \mid \mathcal{S}_t, \texttt{phase}_t) \cdot \delta_{td} + \delta_{td}^2$
20:         $\theta \leftarrow$ update$(\mathcal{L}_\theta, \theta)$
21:     **end for**
22: **end for**
23: **return** $\theta$

---

# D    PSEUDO-CODE FOR FEASIBILITY-AWARE VARIABLE SELECTION

---

**Algorithm 3** Feasibility-aware variable selection

---

**Input:** instance $M$, observation $\mathcal{S}_t = (\mathbf{x}_t, \mathbf{f}_t, obj_t)$, current phase `phase`
**Parameter:** number of seed variables $p$, number of neighboring variables $q$
**Output:** observation with selected variables $\tilde{\mathcal{S}}_t = (\tilde{\mathbf{x}}_t, \mathbf{f}_t, obj_t)$

1: $\tilde{\mathbf{A}} \leftarrow \mathbb{I}(\mathbf{A}_{j,i} \neq 0)_{j=1,\dots,m; i=1,\dots,n}$
2: **if** `phase = 1` **then**
3:     $\tilde{\mathbf{f}}_t \leftarrow \mathbb{I}(f_{t,j} < 0)_{j=1,\dots,m}$
4:     `score_seed` $\leftarrow \tilde{\mathbf{f}}_t^\top \tilde{\mathbf{A}}$                              $\{\tilde{\mathbf{f}}_t \in \mathbb{R}^{m \times 1}, \tilde{\mathbf{A}} \in \mathbb{R}^{m \times n}\}$
5:     `weight` $\leftarrow (\max(\mathrm{abs}(\mathbf{c}^\top)) - \mathrm{abs}(\mathbf{c}^\top) + 1)/\max(\mathrm{abs}(\mathbf{c}^\top))$
6: **else if** `phase = 2` **then**
7:     $\tilde{\mathbf{f}}_t \leftarrow \mathbb{I}(f_{t,j} > 0)_{j=1,\dots,m}$
8:     `score_seed` $\leftarrow \tilde{\mathbf{f}}_t^\top \tilde{\mathbf{A}}$                              $\{\tilde{\mathbf{f}}_t \in \mathbb{R}^{m \times 1}, \tilde{\mathbf{A}} \in \mathbb{R}^{m \times n}\}$
9:     `score_seed` $\leftarrow \max(\text{score\_seed}) - \text{score\_seed} + 1$
10:     `weight` $\leftarrow \mathrm{abs}(\mathbf{c}^\top)/\max(\mathrm{abs}(\mathbf{c}^\top))$
11: **end if**
12: `score_seed` $\leftarrow$ `score_seed` $\odot$ `weight`                    $\{\text{score\_seed} \in \mathbb{R}^{1 \times n}\}$
13: `prob` $\leftarrow$ `score_seed`$/\mathrm{sum}(\text{score\_seed})$                              $\{\text{prob} \in \mathbb{R}^{1 \times n}\}$
14: `indices_seed` $\leftarrow \mathrm{sample}(\text{prob}, p)$          {Sample $p$ seed variables according to `prob`}
15: $\mathbf{g} \leftarrow \mathrm{rowwise\_sum}(\tilde{\mathbf{A}}[:, \text{indices\_seed}])$                              $\{\mathbf{g} \in \mathbb{R}^{m \times 1}\}$
16: `score_neighbor` $\leftarrow \mathbf{g}^\top \tilde{\mathbf{A}}$                    $\{\text{score\_neighbor} \in \mathbb{R}^{1 \times n}\}$
17: `score_neighbor`$[:, \text{indices\_seed}] \leftarrow -1$                    {Prevent to select seed variables}
18: `indices_neighbor` $\leftarrow \mathrm{top}(\text{score\_neighbor}, q)$    {Obtain top $q$ neighboring variables}
19: `changeable` $\leftarrow \mathrm{concatenate}(\text{indices\_seed}, \text{indices\_neighbor})$
20: $\tilde{\mathbf{x}}_t \leftarrow \mathbf{x}_t[\text{changeable}]$                              {Select $\tilde{n}(= p + q)$ variables}
21: $\tilde{\mathcal{S}}_t \leftarrow (\tilde{\mathbf{x}}_t, \mathbf{f}_t, obj_t)$
22: **return** $\tilde{\mathcal{S}}_t$

---

# E    ADDITIONAL PRELIMINARIES

## E.1    PROPERTIES OF ILP

All ILP problems can be transformed into the standard form (Eqs. 1a-1d) (Bertsimas & Tsitsiklis, 1997). Let $\mathbf{a}_i^\top$ denote a row vector of a constraint, $\mathbf{A} = \left(\mathbf{a}_1^\top, \dots, \mathbf{a}_m^\top\right)$, and $\mathbf{b} = (b_1, \dots, b_m)$. An equality constraint $\mathbf{a}_i^\top \mathbf{x} = b_i$ is equivalent to two inequality constraints: $\mathbf{a}_i^\top \mathbf{x} \geq b_i$ and $\mathbf{a}_i^\top \mathbf{x} \leq b_i$. Moreover, $\mathbf{a}_i^\top \mathbf{x} \geq b_i$ is equivalent to $-\mathbf{a}_i^\top \mathbf{x} \leq -b_i$. Maximizing $\mathbf{c}^\top \mathbf{x}$ is equivalent to minimizing $-\mathbf{c}^\top \mathbf{x}$. Thus, we only address the standard form (i.e., minimization) in this study.

ILP is known to be NP-hard due to its integrality constraints (Eq. 1c) (Berthold, 2006). As the number of integer variables increases, the computational cost grows exponentially (Floudas, 1995). LP-relaxation transforms an original ILP problem into an LP one by removing the integrality constraints (Bertsimas & Tsitsiklis, 1997). Although LPs are computationally cheaper to solve, which can be solved in polynomial time (Karmarkar, 1984; Vanderbei, 1998), an LP-feasible solution may be infeasible for the original ILP due to the integrality constraints (Eq. 1c) (Guieu & Chinneck, 1999).

## E.2    START PRIMAL HEURISTICS FOR ILP

Start primal heuristics (Figure 1(b)) do not require an initial ILP-feasible solution. Instead, they typically begin with an LP-feasible solution and attempt to convert it into an ILP-feasible one (Berthold, 2006). Representative methods include diving, feasibility pump (FP), rounding, and relaxation enforced neighborhood search (RENS) (Berthold, 2006; Shoja & Axehill, 2023). Diving methods fix fractional variables in the LP solution to promising integer values and iteratively resolve the LP. FP alternates between two sequences, one LP-feasible and the other ILP-feasible, with the goal of convergence to a feasible ILP solution. Rounding methods attempt to obtain an ILP-feasible solution by rounding fractional LP values up or down. RENS constructs and solves a sub-ILP of the original problem by fixing or tightening bounds of integer variables based on the LP solution.

### E.3 Bipartite graph representation of ILP

Recent studies on E2EPH represent ILP instances as bipartite graphs (Nair et al., 2020; Yoon, 2022; Han et al., 2023; Cantürk et al., 2024; Huang et al., 2024; Liu et al., 2025). In this representation, one set of nodes corresponds to constraints, and the other to decision variables. An edge connects a variable node to a constraint node if and only if the variable appears in the corresponding constraint. For example, in Figure 2(a), variable $x_3$ appears in constraint $\mathbf{a_2}$; thus, the node representing $x_3$ is connected to the node representing $\mathbf{a_2}$ in the bipartite graph.

## F Related work

ML techniques for solving ILP can be broadly categorized into three groups (Bengio et al., 2021). The first group, *learning to configure algorithms*, use ML to optimize the configuration of specific components within ILP solvers. Examples include deciding parameters for promising configurations (Hutter et al., 2011), applying decomposition (Kruber et al., 2017), and selecting scaling methods (Berthold & Hendel, 2021). The second group, *ML alongside optimization algorithms*, integrates ML into ILP solvers to aid key decisions during the optimization, such as cut selection (Tang et al., 2020; Paulus et al., 2022), variable selection (Khalil et al., 2016; Alvarez et al., 2017; Gasse et al., 2019; Gupta et al., 2020; Paulus & Krause, 2023), and node selection in the branch-and-bound (He et al., 2014; Labassi et al., 2022), and neighborhood selection in LNS (Song et al., 2020; Sonnerat et al., 2021; Wu et al., 2021a; Huang et al., 2023; Liu et al., 2024b).

RL-SPH belongs to the third group, *end-to-end learning*, which uses ML to directly learn and predict solutions. This group includes existing E2EPH methods (Nair et al., 2020; Shen et al., 2021; Yoon, 2022; Han et al., 2023; Cantürk et al., 2024; Huang et al., 2024; Liu et al., 2025; Zeng et al., 2024; Geng et al., 2025) as well. PAS (Han et al., 2023) adopts a predefined trust region instead of strictly fixing variables, which can be viewed as the generalization of the fixing strategy (Huang et al., 2024). Recent methods (Huang et al., 2024; Liu et al., 2025) follow this approach as well. Although using trust regions alleviates the risk of infeasibility, these methods still rely on ILP solvers to obtain feasible solutions. Several efforts have been made to solve ILPs via ML without relying on ILP solvers (Zeng et al., 2024; Geng et al., 2025), but none have attained a 100% feasibility rate. Moreover, the aforementioned methods do not support ILP with non-binary integer variables. To the best of our knowledge, RL-SPH is the first E2EPH method that learns to generate feasible solutions independently, offering theoretical feasibility guarantees and empirically achieving a 100% feasibility rate, even for ILP involving non-binary integers.

## G Positioning relative to prior work

RL-SPH is a start primal heuristic designed to quickly achieve feasibility from an initially infeasible ILP solution. As discussed in Appendix F, RL-SPH can also be categorized as an *end-to-end learning*, but it fundamentally differs from existing E2EPH approaches in terms of how feasible solutions are obtained. For example, as illustrated in Figure 1, typical E2EPH methods rapidly approximate a solution and then delegate the remaining optimization—including ensuring feasibility—to an external solver. In this workflow, the solver bears responsibility for producing a feasible solution, typically via its built-in start primal heuristics. Consequently, most prior E2EPH work has evolved under the assumption that feasibility will be handled by the solver rather than by the learned model itself.

This distinction underscores that RL-SPH addresses a more challenging problem than existing E2EPH methods, as it must generate feasible solutions *without* relying on an external solver. Many real-world applications—such as real-time risk assessment (Chen et al., 2023)—require high-quality feasible solutions quickly and repeatedly (Zeng et al., 2024; Cantürk et al., 2024). In such settings, RL-SPH is advantageous because it can quickly deliver feasible solutions independently of a solver. Meanwhile, as observed in prior E2EPH methods (Nair et al., 2020; Huang et al., 2024; Geng et al., 2025), primal heuristics often sacrifice optimality. Therefore, in addition to generating feasible solutions, we further discuss mitigating this drawback within RL-SPH as an important direction for future work in Appendix K.10.

Feasibility is a necessary prerequisite for any subsequent optimization stage. For example, widely used ILP metaheuristics such as local branching (Fischetti & Lodi, 2003; Liu et al., 2022) and large neighborhood search (LNS) (Shaw, 1998; Song et al., 2020; Wu et al., 2021a; Huang et al.,

2023; Liu et al., 2024b) require an initial feasible solution to operate and further improve the incumbent solution. As a result, infeasibility constitutes a significant bottleneck in solving ILP problems. However, finding a feasible solution remains inherently challenging due to the integrality constraints (Eq. 1c), which induce an exponential increase in computational cost with respect to the number of integer variables (Floudas, 1995). Moreover, even well-established primal heuristics typically cannot guarantee feasibility (Shoja & Axehill, 2023). Thus, the ability to reliably generate feasible solutions for general ILPs is crucial. Despite the importance of feasibility, learning-based approaches for directly addressing this problem remain underexplored. Advancing learning-based methods for achieving feasibility establishes a crucial foundation for ML-based stand-alone solvers pursuing optimality, especially given that achieving optimality is recognized as a major challenge in the ML for CO community (Son et al., 2023).

To the best of our knowledge, no learning-based start primal heuristic that independently generates feasible solutions for ILP achieves a 100% feasibility rate (FR). Even when we broaden the scope to E2EPH methods, they still struggle to achieve 100% FR without relying on external solvers. For a more concrete comparison, we compare RL-SPH with recent E2EPH methods in terms of FR, as reported in top-tier conference papers (Zeng et al., 2024; Geng et al., 2025). According to the results reported by Zeng et al., even when the proposed method is combined with SCIP's CompleteSol heuristic, it still fails to obtain 100% feasible solutions on CA. Similarly, Geng et al. have reported the average feasible ratios of 50.8%, 97.1%, and 99.4% on SC, IS, and CA, respectively. Therefore, we consider classical start primal heuristics such as Rounding and FP—both of which achieve higher FR on these benchmarks—to be stronger and more appropriate baselines for our evaluation. To further strengthen our claim, we also present empirical evidence in Appendix J.1 by experimentally comparing RL-SPH with state-of-the-art E2EPH. Furthermore, RL-SPH does not require near-optimal ILP solutions as training labels unlike most existing E2EPH methods, and it achieves approximately 34× faster training time compared to an unsupervised learning–based E2EPH, as detailed in Appendix K.3. This training efficiency makes RL-SPH more practical.

Since existing E2EPH methods and RL-SPH do not operate on the same axis, they offer complementary strengths and can therefore be integrated. For instance, E2EPH methods excel at rapidly approximating high-quality solutions, and this capability can be incorporated into RL-SPH to replace its solution initialization component. The experimental results in Appendix J.1 suggest that RL-SPH has the potential to quickly achieve better feasible solutions when combined with existing E2EPH, making the connection between these two approaches a highly promising direction for future research. In particular, since existing E2EPH methods are largely limited to binary variables, research on learning-based initialization for non-binary integers will be necessary, as discussed in Appendix J.4.

As explained in Section 2.4, RL-SPH resembles LNS in that it explores neighborhoods larger than those used in traditional local search. However, as mentioned earlier, existing learning-based LNS methods require an initially feasible solution and aim to improve the incumbent solution, which positions them in a different stage of the solution process from start primal heuristics. Therefore, they are not direct points of comparison for our study. Nonetheless, insights from the LNS literature could be leveraged to further enhance the behavior of RL-SPH *after* it obtains a feasible solution.

## H  ADDITIONAL EXPLANATION OF REWARD FUNCTION

### H.1  RATIONALE ON REWARD DESIGN

As defined in Section 2.1, ILP is an optimization problem that minimizes an objective value while satisfying a set of constraints: linear constraints, integrality constraints, and variable bounds. When framing ILP within an RL system, the reward mechanism must likewise be designed to reflect this definition. Accordingly, we derived the following fundamental types of rewards: those related to objective values, linear constraints, and variable bounds. The reward for integrality constraints is omitted because they are handled by the action policy (see Section 3.1.1).

According to the above definition, satisfying the set of constraints is a prerequisite for performing optimization. Therefore, rewards related to the set of constraints should take high priority over the objective-value reward. We believe it is intuitively reasonable that the more a bound or linear constraint is violated, the greater the penalty should be. Likewise, a larger reward is appropriate as the objective value improves. From an optimization perspective, improving the feasibility of linear constraints while violating variable bounds is meaningless. For example, in an ILP where variables must take values of either 0 or 1, assigning a value such as –1 is invalid. Thus, satisfying the variable

bounds has the highest priority among the variable bounds, linear constraints, and objective values. The reward structure described in Eq. 4 obeys the mentioned priority. Additionally, to avoid the bound penalty being overly dominated by the constraint reward, we scaled it by the square root of the number of changeable variables. The maximum value of the objective coefficients scales the reward regarding the objective value. This scaling helps stabilize learning by mitigating large reward fluctuations as well.

## H.2 FEASIBILITY REWARD

The feasibility reward $\mathcal{R}_{t,\mathrm{F}}$ is calculated in proportion to the degree of feasibility improvement or deterioration. In Figure 6, the agent violates Constraint-1 while satisfying Constraint-2. If the agent moves closer to the feasible region, it is considered an improvement in feasibility. Conversely, if it moves further away and ends up violating Constraint-2 as well, it is regarded as a deterioration. In this way, rewards are assigned based on how much the agent's actions improve or worsen feasibility. As a result, the agent learns to find feasible solutions to maximize its rewards.

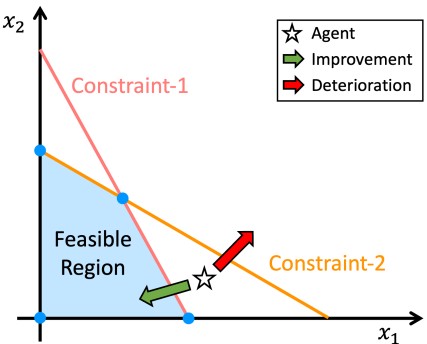

Figure 6: Illustration of feasibility improvement and deterioration.

## H.3 REWARD IN $phase2$

The green circles in Figure 7 represent better feasible solutions whose objective values is $obj_{t+1} < obj_b$ (Case 1 in Eq. 5). We regard the agent's actions that fail to find better feasible solutions than $\mathbf{x}_b$ as incorrect (Cases 2, 3, 4). The red circle indicates a feasible solution worse than the incumbent (Case 2), while all triangles correspond to infeasible solutions (Cases 3, 4).

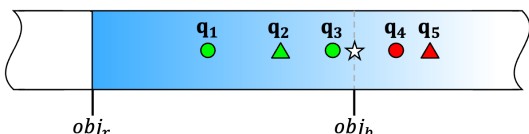

Figure 7: Illustration of reward function in $phase2$. ○: feasible, △: infeasible, green: inside, red: outside, ☆: agent, $obj_r$: the objective value of the LP-feasible solution, $obj_b$: incumbent value.

Let the objective values be $obj_b = -10$, $\mathbf{q_1} = -16$, $\mathbf{q_2} = -13$, $\mathbf{q_3} = -11$, $\mathbf{q_4} = -8$, and $\mathbf{q_5} = -7$, with $\alpha = 2$ and $\max(|\mathbf{c}|) = 1$. For the green circles, rewards in $phase2$ are calculated by the first case in Eq. 5. The rewards for $\mathbf{q_1}$ and $\mathbf{q_3}$ are $\Delta obj_t = \frac{|-16-(-10)|}{1} = 6$ and $\Delta obj_t = \frac{|-11-(-10)|}{1} = 1$, respectively. Since $\mathbf{q_1}$ has the better objective value than $\mathbf{q_3}$, it receives a higher reward. For the red circle, the reward is $-\Delta obj_t \cdot \alpha = -\frac{|-8-(-10)|}{1} \cdot 2 = -4$. For the triangles, the penalties for $\mathbf{q_2}$ and $\mathbf{q_5}$ are $\mathcal{R}_{t,\mathrm{F}}$ and $\mathcal{R}_{t,\mathrm{F}} \cdot \alpha = 2 \cdot \mathcal{R}_{t,\mathrm{F}}$, respectively. The distances from $obj_b$ to $\mathbf{q_2}$ and $\mathbf{q_5}$ are the same (i.e., $|-13 - (-10)| = |-7 - (-10)| = 3$), but the penalty for $\mathbf{q_2}$ is smaller than that for $\mathbf{q_5}$ due to amplifying by $\alpha$.

### H.4 TOWARD-OPTIMAL BIAS

Figure 8 visualizes the agent's potential penalties by $\mathcal{R}_{t,\text{F}}$ in $phase2$. The third and fourth quadrants illustrate the penalties, assuming that $\mathcal{R}_{t,\text{F}}$ is a linear function of the gap from $obj_{t+1}$ to the incumbent value $obj_b$ for simplicity. As illustrated in Figures 8(a) and 8(b), a higher $\alpha$ results in higher penalties for solutions with $obj_{t+1} > obj_b$. By controlling the toward-optimal bias $\alpha$, we can guide the agent to explore promising regions (i.e., $obj_{t+1} < obj_b$) for better feasible solutions rather than making ineffective moves (i.e., $obj_{t+1} > obj_b$), thus increasing its opportunities for learning. Appendix J.5.2 provides experimental results on the toward-optimal bias $\alpha$.

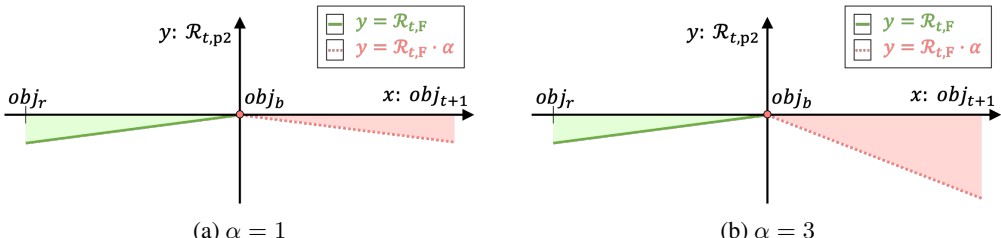

(a) $\alpha = 1$         (b) $\alpha = 3$

Figure 8: Illustration of the potential penalties of $\mathcal{R}_{t,\text{p2}}$ as a function of the objective value $obj_{t+1}$ in $phase2$. The $x$-axis represents $obj_{t+1}$, and the $y$-axis represents $\mathcal{R}_{t,\text{p2}}$, with the origin set at $obj_b$.

## I DETAILS OF EXPERIMENTAL SETUP

### I.1 BENCHMARK DATASETS

Table 7 shows the average sizes of each benchmark dataset used in our experiments. We generated instances for IS, CA, SC, and MVC following the code[1] from (Gasse et al., 2019). For NBI, instances were generated based on the description in (Qi et al., 2021). Table 8 summarizes the parameters used for NBI instance generation. Considering that the ratio of non-zero coefficients $\rho$ in typical LP problems is less than 5% (Hillier & Lieberman, 2015), we set a higher density of 10% to promote more interactions between variables in constraints. According to the default settings of the ILP solver Gurobi (Gurobi Optimization, 2025) and SCIP (Bestuzheva et al., 2021), the lower bound $l_i$ and upper bound $u_i$ for decision variables are set to 0 and $\infty$, respectively.

Table 7: Average sizes of each dataset.

| Dataset | # binary variables | # integer variables | # constraints | Density |
|---|---|---|---|---|
| Independent set (IS) | 1,500 | 0 | 5,941 | 0.13% |
| Combinatorial auction (CA) | 4,000 | 0 | 2,677 | 0.21% |
| Set covering (SC) | 3,000 | 0 | 2,000 | 5% |
| Minimum vertex cover (MVC) | 3,000 | 0 | 11,931 | 0.07% |
| Non-binary integers (NBI) | 0 | 2,000 | 2,000 | 10% |

Table 8: Parameters for non-binary integer instance generation.

| Parameter | Distribution |
|---|---|
| **c** | randint$[-10, 1]$ |
| **A** | randint$[1, 10]$ with density $\rho = 0.1$ |
| **b** | $\mathbf{A}\xi + \epsilon$, where |
| | $\xi_i \sim \text{randint}[1, 10], \forall i = 1, \ldots, n$ and |
| | $\epsilon_j \sim \text{randint}[1, 10], \forall j = 1, \ldots, m$ |
| $l_i$ | $0, \forall i = 1, \ldots, n$ |
| $u_i$ | $\infty, \forall i = 1, \ldots, n$ |

---

[1]https://github.com/ds4dm/learn2branch

## I.2 Evaluation environment

We conducted all evaluations under identical configurations. The evaluation machine is equipped with two AMD EPYC 7302 @ 3.0GHz, 2048GB RAM, and four NVIDIA A100 GPUs. All experiments were performed using a single NVIDIA A100 GPU. The software environment includes PyTorch 1.12.0, Gymnasium 0.29.1, and SCIP 8.1.0.

## I.3 Implementation details

The agent of RL-SPH was trained using the proposed learning algorithm (see Algorithm 2) with RMSprop (learning rate = 1e-4, epsilon = 1e-5, alpha = 0.99, weight decay = 1e-3). The learning rate was linearly decayed over the training epochs. The agent of RL-SPH was trained concurrently on 64 different instances using the parameters for Algorithm 2 and Algorithm 3 as follows: update limit $N = 5000$, total step limit $T_{max} = 2000$, step limit for $phase1$ $T_{stay} = 500$, number of seed variables $p = \lceil \log_2 n \rceil$, and number of neighboring variables $q = \lceil \log_2 n \rceil$. The same hyperparameter configuration was used for training across all datasets. Table 9 shows the training times for the datasets described in Table 7, which were approximately 31, 32, 26, 83, and 23 minutes for IS, CA, SC, MVC, and NBI with $N = 5000$, respectively. Table 9 also shows the size of the GNN trained on each dataset. We compare RL-SPH's training costs with E2EPH methods in Appendix K.3.

We implemented our GNN based on the Transformer code from GitHub[2] (Wu et al., 2021b), with the same configuration. We utilized the positional encoding module from GitHub[3] (Gorishniy et al., 2022). We used the code for PAS (Han et al., 2023) in our experiment, which is available on GitHub[4]. Our RL algorithm is built upon the Actor-Critic implementation in PyTorch[5] (Kostrikov, 2018), modified to be tailored for ILP.

Table 9: Training time and model size of RL-SPH on each dataset.

| Dataset | IS | CA | SC | MVC | NBI |
|---|---|---|---|---|---|
| Training time (min) | 31 | 32 | 26 | 83 | 23 |
| # trainable parameters | 3.9M | 2.7M | 2.4M | 6.2M | 2.4M |

## J Additional experiments

### J.1 Comparison with SOTA E2EPH

We compare RL-SPH with two E2EPH methods: PAS (Han et al., 2023), a representative E2EPH approach, and DiffILO (Geng et al., 2025), the latest E2EPH method. DiffILO has been reported to generate better feasible solutions than other E2EPH methods (Huang et al., 2024; Zeng et al., 2024). The datasets used in this experiment are IS, CA, and SC, shown in Table 7, which are the intersection of the benchmarks used in the DiffILO paper and ours. Since PAS does not provide hyperparameters for the SC benchmark, we exclude it from this experiment. We trained PAS and DiffILO on our datasets using the default configurations and the authors' released code. We utilized the best PAS and DiffILO models obtained during 1,200 epochs of training, and this epoch configuration is among those reported in the DiffILO paper. As shown in Table 10, PAS required an average of 8.7 times more training time, and DiffILO took roughly 34 times longer than RL-SPH. This indicates that both methods were allocated sufficient training time in our experiment.

Table 10: Training times (in minutes) on IS, CA, and SC datasets.

(a) PAS versus RL-SPH.

| Dataset | IS | CA |
|---|---|---|
| PAS | 209 | 343 |
| RL-SPH | 31 | 32 |
| Speedup | **6.7×** | **10.7×** |

(b) DiffILO versus RL-SPH.

| Dataset | IS | CA | SC |
|---|---|---|---|
| DiffILO | 1,002 | 1,177 | 850 |
| RL-SPH | 31 | 32 | 26 |
| Speedup | **32.3×** | **36.8×** | **32.7×** |

---

[2] https://github.com/ucbrise/graphtrans
[3] https://github.com/yandex-research/rtdl-num-embeddings
[4] https://github.com/sribdcn/Predict-and-Search_MILP_method
[5] https://github.com/ikostrikov/pytorch-a2c-ppo-acktr-gail

As shown in Table 11, only RL-SPH and Rounding consistently achieved a 100% FR across all datasets. In contrast, PAS failed to find a feasible solution on IS and CA, aligning with the results presented in the DiffILO paper. Furthermore, DiffILO exhibited 0% FR on CA and SC within the time limit, despite making an average of 574,240 and 919,470 trials of a binomial distribution, respectively. We attribute the FR performance gap between RL-SPH and DiffILO to the prioritization between the objective function and the constraints. DiffILO is trained using a soft-constrained loss function, where a penalty coefficient hyperparameter determines the relative influence of the objective and constraints, meaning that no explicit priority is enforced. In other words, the objective function may dominate the constraints during training, or vice versa. Although the DiffILO study introduced an adaptive penalty strategy, it only partially mitigates the issue and cannot be viewed as a complete solution. Consequently, DiffILO's 0% FR on CA and SC can be attributed to the objective function dominating the constraints within its loss formulation. In contrast, RL-SPH enforces a clear priority between the two: even if the objective value improves, a positive reward (i.e., $+\Delta obj_t$) is not given when feasibility deteriorates (see Eq. 4).

Table 11: Performance comparison on IS, CA, and SC. **Bold** and underline are used to indicate the best and second-best methods among those with 100% FR. *fail* indicates that the metrics cannot be computed because no feasible solution was obtained (i.e., FR = 0%). The time limit is set to 30, 300, and 300 seconds for IS, CA, and SC, respectively.

| Dataset | Metric | Classical heuristics | | Learning-based | | RL-SPH (Ours) | | |
|---|---|---|---|---|---|---|---|---|
| | | FP | Rounding | PAS | DiffILO | w/ Random | w/ LP | w/ DiffILO |
| IS | FR (%) ↑ | 100 | 100 | 0 | 100 | 100 | 100 | 100 |
| | PG (%) ↓ | 34.32±1.29 | 32.24±1.94 | | 0.34±0.50 | 17.87±1.85 | 15.35±1.54 | **0.00**±0.00 |
| | PI ↓ | 11.0±0.4 | 10.8±0.5 | *fail* | 1.1±0.1 | 8.5±0.5 | 7.1±0.4 | **1.0**±0.1 |
| | FT (sec) ↓ | 0.22±0.04 | **0.10**±0.00 | | 0.35±0.06 | 0.27±0.07 | 0.55±0.07 | 0.57±0.12 |
| CA | FR (%) ↑ | 100 | 100 | 0 | 0 | 100 | 100 | 100 |
| | PG (%) ↓ | 22.80±4.66 | 17.16±2.77 | | | 13.04±2.11 | **0.32**±2.26 | 7.84±2.26 |
| | PI ↓ | 77.0±13.6 | 58.6±8.1 | *fail* | *fail* | 71.5±5.6 | **33.3**±4.6 | 52.8±6.2 |
| | FT (sec) ↓ | 10.66±0.70 | 8.03±0.51 | | | **0.41**±0.08 | 8.26±0.72 | 4.65±0.16 |
| SC | FR (%) ↑ | 5 | 100 | | 0 | 100 | 100 | 100 |
| | PG (%) ↓ | 0.00±0.00 | 23.28±4.03 | | | 4.47±7.53 | **0.98**±1.72 | 3.28±7.56 |
| | PI ↓ | 243.6±93.8 | 160.0±17.8 | − | *fail* | 15.4±22.4 | 99.6±8.0 | **12.2**±22.4 |
| | FT (sec) ↓ | 242.62±93.77 | 104.09±16.97 | | | **0.19**±0.07 | 95.43±7.39 | 0.57±0.12 |

To achieve high feasibility comparable to the results reported in the DiffILO study (50.8% on SC and 99.4% on CA), additional hyperparameter tuning for DiffILO would likely be necessary. In our experiment, DiffILO achieved a 100% FR on the IS dataset, whose characteristics closely match those used in the DiffILO study, as shown in Table 12. In contrast, SC differs in the number of variables and constraints, and our CA dataset contains approximately 2.7× more variables and 4.7× more constraints. This indicates that, even within the same problem class, variations in dataset characteristics may necessitate additional tuning for DiffILO to maintain high performance. This claim is further supported by Table 4 in the DiffILO paper, which shows that DiffILO performed dataset-specific tuning on 11 important hyperparameters. In contrast, RL-SPH did not employ any dataset-specific tuning; the same configuration for hyperparameters was used for all datasets, as detailed in Appendix I.3. Therefore, the fact that RL-SPH achieved a 100% FR in all experiments without dataset-specific tuning demonstrates that it can be trained to produce feasible solutions more reliably.

Table 12: Characteristics of datasets used in the DiffILO study and ours.

| Dataset | Study | Number of variables | Number of constraints |
|---|---|---|---|
| IS | DiffILO | 1,500 | 5,943 |
| | Ours | 1,500 | 5,941 |
| CA | DiffILO | 1,500 | 576 |
| | Ours | 4,000 | 2,677 |
| SC | DiffILO | 2,000 | 3,000 |
| | Ours | 3,000 | 2,000 |

Remarkably, the connection of RL-SPH with DiffILO yielded a 100% FR across all datasets and achieved the best performance in both PG and PI on the IS dataset. In addition, RL-SPH was able to convert DiffILO's initially infeasible solutions into feasible ones on CA and SC, while achieving higher PG and PI than RL-SPH(Random) and faster FT than RL-SPH(LP). As shown in Figure 9, the curve for RL-SPH(DiffILO) lies between those of RL-SPH(Random) and RL-SPH(LP) on CA and SC. This indicates that the initial solutions provided by DiffILO offer a better starting point than random initialization, but a weaker one compared to LP-relaxation. Since learning-based models offer speed advantages when approximating initial solutions, a refined approximation could potentially lead to superior performance compared to LP-based initialization, as demonstrated on IS. This finding supports the claim in Appendix G that existing E2EPH methods and RL-SPH complement one another. Meanwhile, PAS and DiffILO do not support non-binary integer variables, so we were unable to conduct experiments on NBI. This limitation highlights the need for further research on solution approximation methods for non-binary integer variables. Interestingly, RL-SPH(DiffILO) was trained with LP initialization rather than with initial solutions generated by DiffILO, indicating that RL-SPH generalizes well to a different initialization method.

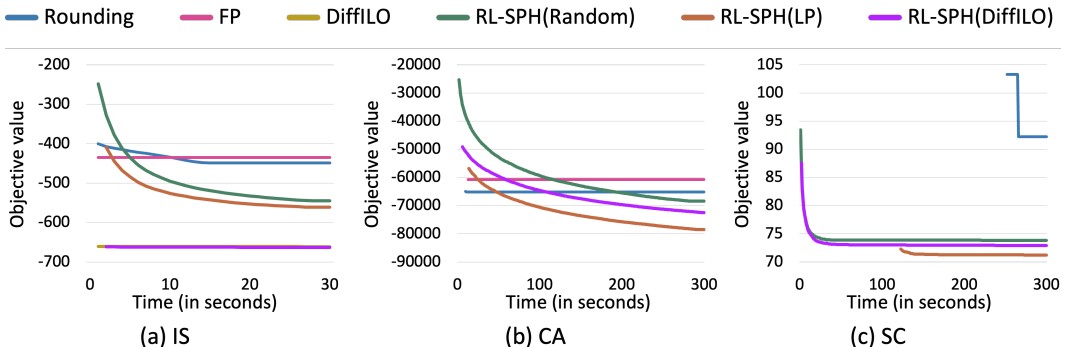

Figure 9: Illustration of the objective values of the compared methods over time (in seconds). Lines indicate valid averaged incumbent objective values, which are computed only when an incumbent objective value exists for all instances at a given time point. Cases with FR = 0% are not displayed.

## J.2 EVALUATION OF GENERALIZATION

### J.2.1 DISTRIBUTION CHANGES

We also conducted an experiment to evaluate the robustness of RL-SPH to changes in distribution. As described in Section 4.1.1, the IS dataset was generated using the Barabási–Albert random graph model (BA). BA generates scale-free networks that follow a power-law degree distribution. It captures the preferential attachment mechanism observed in many real-world networks, such as citation networks and social networks. In contrast, Erdős–Rényi random graph model (ER) constructs a graph by starting with $u$ nodes and connecting each pair of nodes with probability $p$ (Albert & Barabási, 2002). This process yields a graph where the expected number of edges is approximately $p \cdot u(u-1)/2$, with edges distributed uniformly at random. Therefore, graphs generated by ER and BA have different degree distributions.

Table 13 shows that RL-SPH, trained on IS instances generated by BA, produced feasible solutions with a 100% FR on test instances generated by ER. Furthermore, the results demonstrate that RL-SPH rapidly achieved higher-quality solutions than Rounding, which is the strongest baseline. Therefore, RL-SPH robustly finds high-quality feasible solutions under distribution changes.

Table 13: Generalization performance of RL-SPH under distribution changes.

| Test dataset | Metric | Rounding | RL-SPH trained on IS (BA) |
|---|---|---|---|
| IS (ER) | FR (%) ↑ | 100 | 100 |
| | PG (%) ↓ | 7.02±2.39 | **0.00**±0.00 |
| | PI ↓ | 2.7±0.6 | **2.4**±0.2 |

### J.2.2 SIZE CHANGES

CO research on ML for large-scale ILP is crucial. Recently, there have been studies on ML-based heuristics for solving large-scale ILP (Ye et al., 2023; 2024). These works have proposed techniques that utilize a graph partitioning algorithm (Tsourakakis et al., 2014) to divide large-scale ILP instances into smaller subproblems for optimization. Their experimental results showed that breaking a large problem into smaller ones can effectively address the challenges of large-scale ILP. This finding suggests that, even if RL-SPH may not directly handle large-scale ILP, it has the potential to solve them by decomposing the original large-scale problem into smaller subproblems. Therefore, we believe the datasets used in our experiments are sufficient for evaluating our method. Nevertheless, to more thoroughly assess its robustness, we experimented on scalability, as follows.

Table 14 shows that RL-SPH trained on SC ($n = 3,000$) successfully generated feasible solutions for larger test instances. Remarkably, RL-SPH achieved a 100% FR even on test instances with more than three times the number of variables as the training instances. Among the heuristics, we selected Rounding as the baseline for comparison, as it was the only method in the main experiments to achieve a 100% FR (see Table 1). RL-SPH consistently found higher-quality solutions than Rounding, indicating that a model trained on smaller instances can generalize effectively to larger ones.

Table 14: Generalization performance of RL-SPH trained on SC ($n = 3,000$) under size changes.

| Metric | SC ($n = 6,000$) | | SC ($n = 10,000$) | |
|---|---|---|---|---|
| | Rounding | **RL-SPH** | Rounding | **RL-SPH** |
| FR (%) ↑ | 100 | 100 | 100 | 100 |
| PG (%) ↓ | 21.47±3.04 | **0.00**±0.00 | 20.74±3.57 | **0.00**±0.00 |
| PI ↓ | 415.3±26.4 | **237.1**±14.8 | 548.2±31.5 | **372.8**±20.8 |

### J.2.3 DENSITY CHANGES

Table 15 shows that RL-SPH, trained on SC with the density $\rho = 5\%$, consistently generated feasible solutions across test instances with varying densities. Specifically, RL-SPH achieved a 100% FR on both the sparser dataset ($\rho = 1\%$) and the denser one ($\rho = 10\%$). The results demonstrate that RL-SPH quickly obtained higher-quality solutions than Rounding. This indicates that RL-SPH is robust to variations in instance density.

Table 15: Generalization performance of RL-SPH trained on SC ($\rho = 5\%$) under density changes.

| Metric | SC ($\rho = 1\%$) | | SC ($\rho = 10\%$) | |
|---|---|---|---|---|
| | Rounding | **RL-SPH** | Rounding | **RL-SPH** |
| FR (%) ↑ | 100 | 100 | 100 | 100 |
| PG (%) ↓ | 23.77±2.33 | **0.00**±0.00 | 20.73±4.47 | **0.00**±0.00 |
| PI ↓ | 80.8±11.2 | **43.6**±7.4 | 233.4±37.7 | **113.2**±8.7 |

### J.3 EVALUATION OF RL-SPH COMBINED WITH A COMMERCIAL SOLVER

We conducted an additional experiment by combining RL-SPH with the commercial solver Gurobi. This experiment followed the same protocol described in Section 4.3. If RL-SPH fails to find a feasible solution within the time limit, the run would be recorded as a failure, and the same applies in Section 4.3. To ensure a fair comparison with SCIP, Gurobi was configured to use a single thread as done in the previous work (Han et al., 2023), with all other parameters kept at their default settings. Table 16 presents the results on NBI, where RL-SPH+Gurobi denotes RL-SPH combined with Gurobi. The experimental results show that RL-SPH converged to $BKS$ faster when combined with Gurobi.

Table 16: Solving performance of RL-SPH combined with ILP solvers. In all cases, FR is 100%.

| Dataset | Metric | SCIP | Gurobi | RL-SPH+SCIP | **RL-SPH+Gurobi** |
|---|---|---|---|---|---|
| NBI | PG (%) ↓ | 0.25±0.03 | 0.03±0.02 | 0.24±0.04 | **0.03**±0.03 |
| | PI ↓ | 11.5±2.3 | 5.2±1.0 | 4.9±0.9 | **2.4**±0.5 |

### J.4 RANDOM INITIALIZATION ON NBI

In this section, we examine whether RL-SPH can still find feasible solutions for NBI with the random initialization introduced in Appendix C. As shown in Table 17, RL-SPH with random initialization

achieves a 100% FR, indicating that RL-SPH can be trained to obtain feasible solutions from the randomly generated initial point even on NBI. However, the solution quality obtained from random initialization is significantly worse than that from LP initialization. We conjecture that this stems from the unbounded range of the integer variables, which allows them to take larger values. The observation that Random(avg), which assigns values greater than 1, yields lower PG and PI than Random(1) supports this explanation. This gap in solution quality is therefore attributable to the initialization strategy. Given that most existing E2EPH studies are limited to binary problems, this underscores the need for future research on learning-based initialization methods that are better tailored to general ILPs while being faster than LP.

Table 17: The performance of RL-SPH on NBI under different solution initializations. Random(1) assigns the value 1 to randomly selected variables, identical to the method described in Appendix C. Random(avg) assigns the average of the non-zero variable values from the previous instance; apart from that, it works the same as Random(1). During training, the average value was calculated as 4, and Random(avg) uses this value for assignment in the test phase. The time limit is set to 50 seconds.

| Dataset | Initialization type | FR (%) ↑ | PG (%) ↓ | PI ↓ |
|---------|---------------------|----------|----------|------|
| | LP | 100 | **0.00**±0.00 | **10.8**±0.8 |
| NBI | Random(1) | 100 | 30.15±3.38 | 19.8±1.3 |
| | Random(avg) | 100 | 27.79±2.17 | 17.6±0.9 |

### J.5 HYPERPARAMETER ANALYSIS

#### J.5.1 EFFECT OF THE NEIGHBOR VARIABLE RATIO

Table 18 presents the performance of RL-SPH under different ratios of neighboring variables among the changeable variables (i.e., $q/\tilde{n}$). Regardless of the ratio values, RL-SPH successfully finds a feasible solution in all experimental settings. In IS and MVC, higher neighboring variable ratios enable RL-SPH to reach higher-quality feasible solutions more quickly. These results suggest that IS and MVC exhibit stronger interactions among neighboring variables compared with the other datasets. As shown in Table 19, IS and MVC have relatively high variable degrees compared to their constraint degrees (i.e., a higher degree ratio). Consequently, when the agent modifies the value of a seed variable to improve feasibility, it must also account for the many constraints involving that variable, along with the other variables that appear in those constraints. Thus, for datasets with a large degree ratio, a larger neighboring variable ratio is preferable, as it allows the agent to capture interactions among more neighboring variables.

Table 18: The performance of RL-SPH under different ratios of neighboring variables. #win measures the number of test instances in which a method reaches the best value among all compared methods. The time limit is set to 150 seconds for SC and 50 seconds for all other datasets.

| Dataset | Neighbor variable ratio | FR (%) ↑ | PG (%) ↓ | PI ↓ | #win ↑ |
|---------|-------------------------|----------|----------|------|--------|
| | 0.3 | 100 | 8.22±1.70 | 6.9±0.8 | 0 |
| IS | 0.5 | 100 | 5.86±1.94 | 5.7±0.9 | 0 |
| | 0.7 | 100 | **0.00**±0.00 | **2.8**±0.2 | **100** |
| | 0.3 | 100 | **0.99**±2.08 | **11.9**±1.2 | **60** |
| CA | 0.5 | 100 | 1.57±2.11 | 12.1±1.2 | 40 |
| | 0.7 | 100 | 8.84±3.38 | 15.0±1.6 | 0 |
| | 0.3 | 100 | 2.15±2.24 | 97.6±6.7 | 39 |
| SC | 0.5 | 100 | **1.59**±2.10 | 98.2±7.4 | **52** |
| | 0.7 | 100 | 2.55±2.37 | **97.2**±7.4 | 31 |
| | 0.3 | 100 | 7.49±0.94 | 7.8±0.4 | 0 |
| MVC | 0.5 | 100 | 6.90±0.97 | 7.6±0.4 | 0 |
| | 0.7 | 100 | **0.00**±0.00 | **4.8**±0.2 | **100** |
| | 0.3 | 100 | **0.03**±0.04 | 10.9±0.7 | **59** |
| NBI | 0.5 | 100 | 0.05±0.06 | **10.7**±0.6 | 32 |
| | 0.7 | 100 | 0.09±0.06 | 10.9±0.8 | 10 |

Table 19: Average degree of constraints and variables.

| Dataset | IS | CA | SC | MVC | NBI |
|---------|------|------|--------|------|--------|
| Degree of constraints $d_c$ | 2.00 | 8.31 | 150.00 | 2.00 | 198.98 |
| Degree of variables $d_v$ | 7.94 | 5.56 | 100.00 | 7.96 | 198.98 |
| Degree ratio $d_v/d_c$ | 3.96 | 0.67 | 0.67 | 3.98 | 1.00 |

### J.5.2 EFFECT OF THE TOWARD-OPTIMAL BIAS

Table 20 shows the performance of RL-SPH under different values of the toward-optimal bias $\alpha$. With a toward-optimal bias (i.e., $\alpha > 1$), RL-SPH achieves better PG, PI, and #win than the value obtained with $\alpha = 1$. In contrast, RL-SPH without such bias (i.e., $\alpha = 1$) consistently attains the lowest #win across all datasets. These results demonstrate that injecting a toward-optimal bias helps the agent reach higher-quality solutions, highlighting the importance of guiding the agent's search during training. Increasing the bias from $\alpha = 2$ to $\alpha = 5$ does not necessarily yield further improvements. Notably, in SC, RL-SPH($\alpha = 5$) achieves a PG of 2.52%, which is even higher than the value obtained with $\alpha = 1$. Given that the model also exhibits a noticeably high value of #viol_2 in SC, we conjecture that the model becomes overly focused on pursuing a lower objective value ($obj_{t+1} < obj_b$), which in turn leads to more aggressive attempts that may even violate constraints. Therefore, it is advisable to choose $\alpha$ to be greater than 1 but not excessively large.

Table 20: The performance of RL-SPH under different toward-optimal bias values. #viol_2 is the sum of variable bound violations and linear constraint violations in $phase2$. The time limit is set to 150 seconds for SC and 50 seconds for all other datasets.

| Dataset | Toward-optimal bias | FR (%) ↑ | PG (%) ↓ | PI ↓ | #win ↑ | #viol_2 |
|---------|---------------------|----------|-----------|---------|--------|---------|
| IS | $\alpha = 1$ | 100 | 3.41±2.10 | 7.2±1.0 | 9 | 1.0±0.1 |
|    | $\alpha = 2$ | 100 | 0.94±1.29 | 3.4±0.6 | 45 | 0.3±0.1 |
|    | $\alpha = 5$ | 100 | **0.85**±1.20 | **3.3**±0.6 | **53** | 0.4±0.1 |
| CA | $\alpha = 1$ | 100 | 3.22±3.00 | 12.5±1.3 | 17 | 2.2±0.5 |
|    | $\alpha = 2$ | 100 | **0.88**±1.64 | **11.8**±1.1 | **57** | 2.1±0.4 |
|    | $\alpha = 5$ | 100 | 2.49±2.58 | 12.4±1.2 | 26 | 3.1±0.5 |
| SC | $\alpha = 1$ | 100 | 2.39±2.13 | 98.2±6.8 | 32 | 4.0±4.7 |
|    | $\alpha = 2$ | 100 | **1.74**±2.22 | 98.4±7.2 | **50** | 4.7±6.3 |
|    | $\alpha = 5$ | 100 | 2.52±2.41 | **97.8**±7.2 | 33 | 11.2±17.9 |
| MVC | $\alpha = 1$ | 100 | 14.18±0.90 | 8.8±0.4 | 0 | 1.7±0.2 |
|     | $\alpha = 2$ | 100 | **0.43**±0.69 | **4.6**±0.3 | **60** | 0.8±0.1 |
|     | $\alpha = 5$ | 100 | 0.67±0.83 | 4.7±0.4 | 41 | 1.1±0.1 |
| NBI | $\alpha = 1$ | 100 | 0.24±0.11 | 10.9±0.6 | 1 | 1.4±0.3 |
|     | $\alpha = 2$ | 100 | 0.06±0.07 | **10.7**±0.6 | 29 | 2.7±0.6 |
|     | $\alpha = 5$ | 100 | **0.01**±0.03 | 10.8±0.7 | **71** | 4.6±0.7 |

## J.6 QUALITATIVE ANALYSIS

### J.6.1 BEHAVIOR OF THE TWO-PHASE REWARD SYSTEM

To verify that the proposed two-phase reward system operates as intended, we perform an additional analysis. Figure 10 illustrates the optimization trajectory of RL-SPH over 100 search steps on instances randomly sampled from each dataset. As described in Section 3.1.3, the primary goal of $phase1$ is to repair infeasible solutions and obtain the first feasible one. Across all datasets, the initial solutions violate a large number of constraints. For example, the initial solution for MVC violates approximately 3,000 constraints. RL-SPH gradually reduces the number of violated constraints and eventually satisfies all of them, thereby discovering the first feasible solution.

According to the policy defined in Section 3.1.3, $phase2$ begins once the first feasible solution is found. Since the goal of $phase2$ is to improve the incumbent objective value $obj_b$, the agent searches for higher-quality feasible solutions within its neighborhood (see Algorithm 1). As shown in Figure 10, after transitioning into $phase2$, the agent takes moves that preserve feasibility while improving $obj_b$. Across all datasets, the RL-SPH agent exhibits a consistent behavioral pattern, providing evidence that it operates in accordance with the intended design of our reward system.

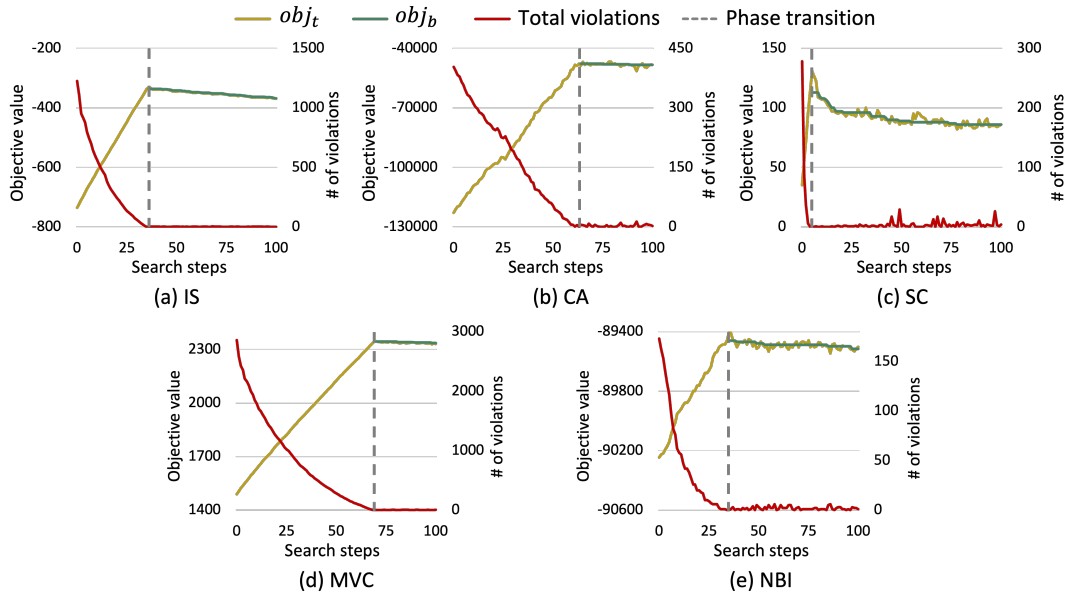

Figure 10: Illustration of RL-SPH's optimization trajectory over search steps, using one randomly sampled instance from each dataset. The total number of violations is computed as the sum of variable bound violations and linear constraint violations. The period before the phase transition corresponds to $phase1$, and the period after the transition corresponds to $phase2$.

### J.6.2 LONG-RANGE VARIABLE DEPENDENCIES

To examine whether the proposed Transformer architecture learns long-range dependencies among variables in ILP problems represented as bipartite graphs, we conduct a case study showing attention weights between distant variables. Figure 11 shows a snapshot captured during the RL-SPH's optimization of a randomly sampled instance from the IS dataset, where interactions among neighboring variables are important as discussed in Section J.5.1. Figure 11(a) reports the shortest-path hop distances between variable nodes that are fed into the Transformer of RL-SPH, while Figure 11(b) depicts the attention weights computed among these variable nodes. Although the reward context is also included as input nodes in practice, it is omitted here for consistency with Figure 11(a).

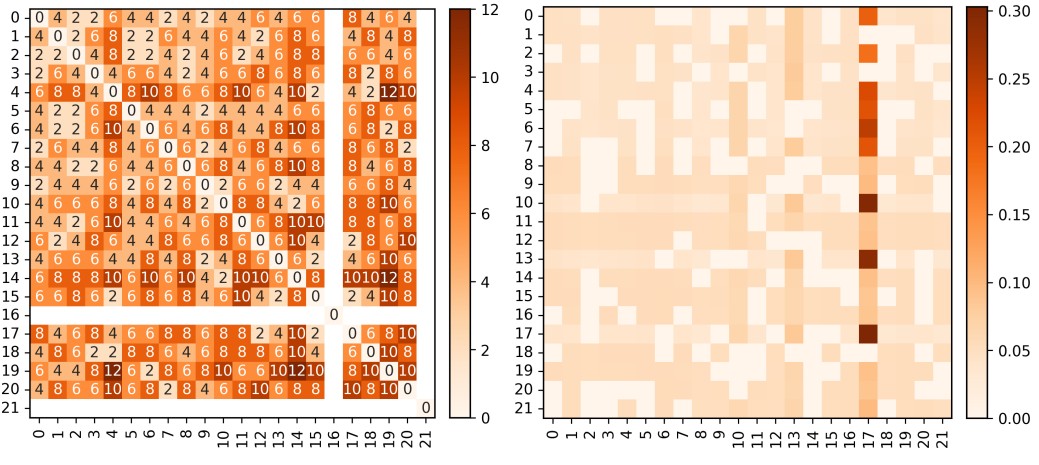

(a) Heatmap of shortest-path hop counts.  (b) Attention map from our Transformer.

Figure 11: Heatmap of shortest-path hop counts and attention map from our Transformer. This example is randomly sampled from the IS test set. The attention map is taken from the final Transformer layer. The x-axis and y-axis correspond to variable nodes. In (a), blank cells in the heatmap indicate that there is no path between the two nodes.

As shown in Figure 11(a), variable-node 10 is eight hops away from variable-node 17, yet it receives a larger attention weight than several nearer nodes. This behavior aligns with observations reported in graph transformer literature (Wu et al., 2021b) and suggests that the proposed Transformer can effectively capture long-range dependencies among decision variables in ILP.

### J.7 Training curves

Figure 12 plots the rewards that RL-SPH receives over time during training on each dataset. Since the reward mechanism of RL-SPH consists of $phase1$ and $phase2$, the reward magnitude periodically fluctuates whenever the agent transitions to $phase2$ or begins $phase1$ of the next instance. As described in Appendix, I.3, the agent stays in $phase1$ for one quarter of the total step limit allocated to each instance, which indicates that the period with relatively higher rewards in Figure 12 corresponds to $phase1$. Except for MVC, the training process converges within about 10 minutes and consistently exhibits a repeating and stable pattern.

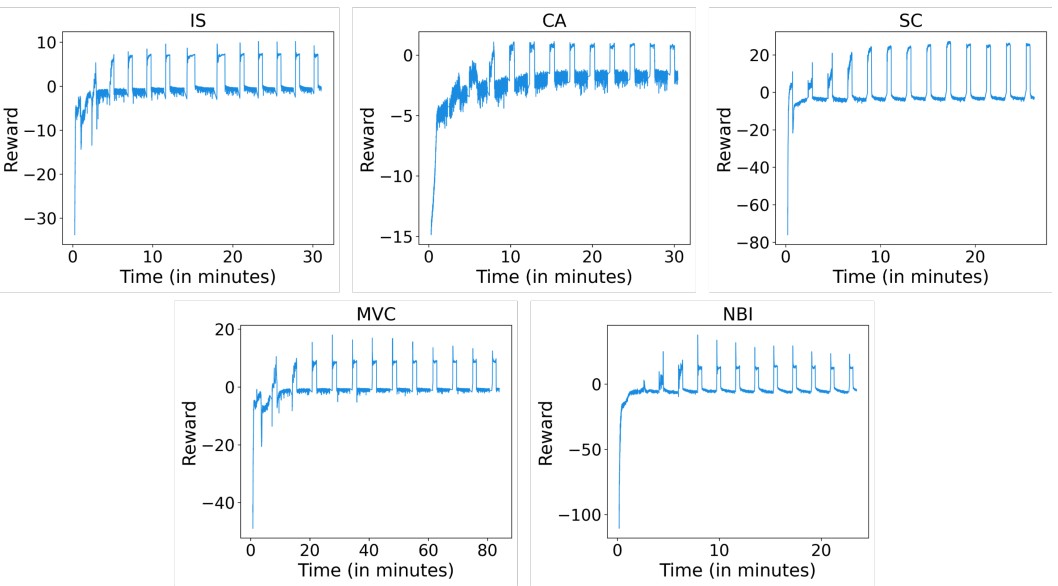

Figure 12: The training curves of RL-SPH on each dataset.

## K    Additional discussion

### K.1    Feasibility as a core focus of this work

In this work, our primary focus is on finding feasible solutions from an infeasible start point, rather than optimizing objective values. By definition in Section 2.1, an optimal solution is *feasible* and achieves the lowest objective value. Even if a solution yields the lowest objective value, it is considered invalid if it violates even a single constraint. Therefore, achieving feasibility is a crucial prerequisite for optimality. Furthermore, reaching optimality is recognized as a major challenge in the ML for CO community (Son et al., 2023), which underscores the importance of guaranteeing feasibility in ILP.

Despite this importance, feasibility is not typically guaranteed by primal heuristics (Shoja & Axehill, 2023), including E2EPH methods (Nair et al., 2020; Shen et al., 2021; Yoon, 2022; Han et al., 2023; Cantürk et al., 2024; Huang et al., 2024; Liu et al., 2025; Zeng et al., 2024; Geng et al., 2025). As discussed in Appendix F and G, to the best of our knowledge, no existing E2EPH has achieved a 100% feasibility rate. These observations suggest that *feasibility rate* (FR) should be regarded as a fundamental evaluation metric in this research area. Therefore, we argue that feasibility should be prioritized and that methods achieving higher FR should be regarded as stronger baselines. From this perspective, we consider Rounding to be a stronger baseline than other primal heuristics, including the E2EPH methods.

## K.2 Time complexity comparison

As shown in Appendix B, RL-SPH can obtain the first feasible solution for ILP from an infeasible starting point in polynomial time. We compare the time complexity of RL-SPH against commonly used start primal heuristics (i.e., Diving, RENS, Rounding, and FP). They typically do not guarantee finding a feasible solution (Shoja & Axehill, 2023), making it difficult to define their exact time complexity for obtaining feasible solutions. Therefore, instead of providing precise time complexities for each heuristic, we offer an intuitive comparison based on their general characteristics.

Since each heuristic requires obtaining an LP-feasible solution, $T_{LP}$ is the lower bound of their complexity. Moreover, Diving and FP require solving LPs repeatedly to obtain a feasible solution, and RENS requires solving a sub-ILP problem (i.e., NP-hard). Given that RL-SPH operates in $O(T_{LP})$ (see Appendix B), RL-SPH is at least comparable to Rounding ($O(T_{LP})$) and more efficient than Diving&FP ($O(k \cdot T_{LP})$) and RENS ($O(T_{sub-ILP})$), where $k$ is the number of iterations.

## K.3 Training costs

Most studies on E2EPH are based on supervised learning models (Nair et al., 2020; Yoon, 2022; Han et al., 2023; Huang et al., 2024; Cantürk et al., 2024; Zeng et al., 2024). Obtaining training labels for ILP (e.g., near-optimal solutions) requires a considerable amount of time. For example, such labels are obtained by running Gurobi with a time limit of 3,600 seconds for a single instance (Han et al., 2023; Huang et al., 2024). Considering that training requires on the order of hundreds of instances, this constitutes a substantial computational cost. This labeling process becomes even more computationally expensive for larger instances, limiting the practicality for NP-hard problems (Cantürk et al., 2024). Moreover, training for the supervised learning models usually converges within 5 hours (Huang et al., 2024).

In contrast, our RL-SPH does not require near-optimal feasible solutions for training. In addition, training on IS took approximately 31 minutes, as reported in Section 4.1.2. All datasets required less than an hour to train RL-SPH, except MVC with 83 minutes, as shown in Table 9. When compared to a representative supervised learning method (Han et al., 2023) in terms of training time, RL-SPH achieved an average speedup of 8.7× on the same datasets, as shown in Table 10(a). Consequently, RL-SPH incurs lower training cost than supervised learning methods, as it requires no labeling process and less training time. For an unsupervised method (Geng et al., 2025), training on our datasets took about 16 hours, whereas RL-SPH was roughly 34× faster on the same datasets, as shown in Table 10(b). This substantial reduction in training time underscores the practicality of RL-SPH.

## K.4 Training efficiency

Our feasibility-aware search strategy enables efficient training. This strategy allows the agent to focus on variables that are more likely to improve feasibility by filtering out uninformative inputs. In a real-world analogy, this is akin to preparing for a final exam by focusing on the topics the professor highlighted during lectures, rather than studying an entire 1000-page textbook including bibliography and index. The former one enables targeted learning, which could result in improved exam outcomes. In this sense, this learning strategy can be viewed as a higher-level attention mechanism—operating at the level of pages—compared to one that focuses on important words within a page. Likewise, our search strategy narrows the broad observation space to more informative regions, enhancing learning efficiency through stronger reward signals. The insights from this learning strategy may also be useful in other domains, such as RL for routing optimization, which requires several days of training (Son et al., 2024).

## K.5 Novelty of the feasibility-aware search strategy

We received the following request from a reviewer: *"Authors should clarify whether it draws from classical methods or is newly designed, what its computational cost is, whether it is a core contribution, and whether there is room for further optimization."*. The feasibility-aware search strategy was newly designed and not adapted from prior techniques. We are not aware of classical methods that resemble our feasibility-aware search strategy. Although the concept of exploring a neighborhood is borrowed from local search as described in Section 2.4, the variable selection policy itself was newly designed.

The computational complexity of Algorithm 1 is determined by three components: variable selection, Transformer processing, and observation computation. For variable selection, multiplying the feasibility state vector $\tilde{\mathbf{f}}_t \in \mathbb{R}^{m \times 1}$ with the coefficient matrix $\tilde{\mathbf{A}} \in \mathbb{R}^{m \times n}$ requires $O(mn)$ operations. Processing $\tilde{n}^2 + 3$ input tokens with the Transformer incurs a cost of $O(\tilde{n}^2)$. For the observation update, only the changeable variables need to be used for computation, which results in $O(m\tilde{n})$. Therefore, the overall computational complexity of Algorithm 1 is $O(mn + \tilde{n}^2)$.

We consider the feasibility-aware search strategy a key contribution, as it offers insight into how the search space can be reduced to enable more efficient learning for the RL agent, as discussed in Appendix K.4. We believe that the proposed search strategy represents only one possible way to reduce the search space, and that further optimization is certainly possible. For example, variable selection currently takes $O(mn)$, which becomes quadratic when $m = n$. Reducing this cost remains an interesting direction for future work.

### K.6   TRINITY AS THE KEY ENABLER

We received the following question from one of the reviewers: *"What is the key enabler of its success: the reward design, the Transformer architecture, the feasibility-aware search strategy, or their combination?"*. Since none of the three components can be removed without compromising the reported performance, the key enabler is the combination of all three. The ablation studies in Section 4.5 show that RL-SPH struggles to find a feasible solution quickly without our search strategy, and RL-SPH achieves the best results with all components of our GNN, supporting this claim.

These three components interact closely with one another, and each component increases the overall complexity of the system.

- **The reward design**: A poorly designed reward function may introduce loopholes that the agent exploits undesirably. For example, the agent may remain stationary to avoid penalties. Additionally, the objective function may outweigh the constraints, leading to violations. To ensure appropriate behavior, the reward function must be carefully crafted (see Appendix H.1), and the GNN must be designed to be context-aware (see RC in Table 6).
- **The feasibility-aware search strategy**: Even with a well-designed reward function and GNNs, introducing irrelevant inputs can slow down training convergence. Even if training succeeds, noisy inputs can degrade solving performance. Consequently, narrowing the search space facilitates more efficient learning and enhances solving effectiveness (see Appendix K.4 and Table 5).
- **The Transformer architecture**: After the variable selection step, the full bipartite graph is no longer available to the model. It means that some bridge nodes are lost and certain reachability paths disappear, as shown in blank cells in Figure 11(a). Consequently, an MPNN-based GNN may struggle to learn effective representations for such isolated variable nodes. Thus, the GNN must be designed to effectively capture long-range dependencies.

Given their tight interaction, the combination of these three components is the key enabler of RL-SPH's success. At the same time, we acknowledge that better designs for each individual component may certainly exist and represent promising avenues for future research.

### K.7   POTENTIAL BOTTLENECKS AT EXTREME SCALES

While our search strategy alleviates the scalability issue by passing only $\tilde{n} \ (= 2 \lceil \log_2 n \rceil)$ inputs to the Transformer, the quadratic complexity of the Transformer's attention mechanism could become a bottleneck for extremely large-scale problems. For such cases, it would be beneficial to consider attention architectures specifically designed for lower computational complexity, such as RetNet (Sun et al., 2023) or other variants.

Even if the quadratic attention in the Transformer is resolved, the variable selection may become a new bottleneck due to its quadratic complexity when $m = n$ (see Appendix K.5). Addressing this issue will likewise be an important direction for future work.

Furthermore, if the number of constraints becomes extremely large, projecting the structural information into the hidden size may incur substantial computational cost. This issue could be addressed by explicitly introducing constraint nodes as part of the Transformer input nodes, rather than compressing

all structural information into a single projected vector. We conducted preliminary experiments to explore this direction and observed promising performance, although more thorough evaluation is certainly required.

### K.8 HANDLING NONLINEAR CONSTRAINTS

Since our study is scoped to *Integer Linear Programming* (ILP), we did not consider explicitly designing our reward system to handle nonlinear constraints. Nonetheless, we suspect that the reward system might also work in the presence of nonlinearities because the reward is computed based on the change from the previous timestep, regardless of the type of constraints. However, this has not yet been tested and remains to be explored further.

### K.9 CHANGES IN MOVEMENT MAGNITUDE

Having a larger neighborhood is effective in enhancing search performance (Shaw, 1998). The size of the neighborhood is associated not only with the number of neighboring variables but also with the magnitude of movement. Table 21 summarizes our preliminary experiments on the effect of varying movement magnitude on search performance, where $\Delta_x$ denotes the maximum movement magnitude. The solution search with $\Delta_x = 5$ led to the highest #win, which leverages the largest action set in this experiment. These results suggest that expanding the movement range enhances the ability to reach better objective values. Despite this finding, we set $\Delta_x = 1$ in our main experiments because our focus is on feasibility rather than enhancing optimality.

To allow larger movement than $\Delta_x = 1$, RL-SPH can be easily extended by expanding the action space. For example, if we want $\Delta_x = 2$, we can expand the action space to {-2, -1, 0, +1, +2}, which the agent chooses from. The observation from the preliminary experiments highlights a promising direction for future research to improve solution optimality within RL-SPH (e.g., a systematic method for configuring movement range variations), which we discuss further in Appendix K.10.

As discussed in Section K.1, our emphasis is on feasibility rather than optimality. Although feasibility is our primary focus, we note that RL-SPH also achieves strong solution quality: even with $\Delta_x = 1$, it outperforms four classical start primal heuristics and the SOTA E2EPH (Geng et al., 2025). Thus, exploring larger action magnitudes would not affect our main conclusions.

Table 21: Performance by neighborhood sizes on the NBI dataset with 100 variables and 50 constraints. 300-second time limit was applied to all cases.

| Max Magnitude | FR (%) ↑ | PG (%) ↓ | #win ↑ |
|---|---|---|---|
| $\Delta_x = 1$ | 100 | 0.36 | 21 |
| $\Delta_x = 3$ | 100 | 0.31 | 40 |
| $\Delta_x = 5$ | 100 | 0.31 | **50** |

### K.10 PROMISING DIRECTIONS TO IMPROVE OPTIMALITY

To further improve the optimality within RL-SPH, several research directions can be explored. First, a promising avenue is the dynamic determination of the action space. The size of the neighborhood is enlarged with the magnitude of movement. Since having a larger neighborhood enhances search performance (Shaw, 1998), a larger movement magnitude could improve the objective value. Since the most suitable action space may depend on the agent's current state, dynamically adapting the action space could be more effective than using a static one. For instance, when the agent is trapped in a deeper local optimum, a larger movement magnitude may be required to escape.

Second, integrating metaheuristics (Hussain et al., 2019), high-level strategies that guide the search process, could further enhance RL-SPH. For example, Iterated Local Search (ILS) (Lourenço et al., 2003) introduces perturbations to escape local optima during the search. In this context, RL-SPH could be extended to detect whether it is stuck in a local optimum and learn how to apply perturbations (both in direction and magnitude) to improve search performance.

Third, RL-SPH could be extended to learn how to select promising neighborhoods. Prior studies on Large Neighborhood Search (LNS) have demonstrated the effectiveness of learning-based neigh-

borhood selection policies (Huang et al., 2023). Building on this line of work, future research could explore how RL-SPH can learn to identify and exploit high-quality neighborhoods.

Finally, we emphasize that optimality cannot be pursued without first ensuring feasibility, which is a crucial prerequisite. Thus, our work serves as a foundation that opens the door to a wide range of future studies aimed at improving solution quality.

