# OpenReview forum: "RL-SPH: Learning to Achieve Feasible Solutions for Integer Linear Programs"
_ICLR.cc/2026/Conference — Submitted to ICLR 2026_

### Official Review · Reviewer_Hj6o · 2025-10-29

**Soundness:** 2
**Presentation:** 3
**Contribution:** 3
**Rating:** 4
**Confidence:** 4

**Summary:**

This paper proposes RL-SPH, a two-phase reinforcement-learning heuristic that aims to produce feasible solutions for ILPs without relying on solver-side repair. The method separates the search into a feasibility-first stage and a quality-improvement stage, uses a Transformer-GNN backbone to encode problem structure and current solution state, and restricts actions to small integer moves over a selected subset of variables. Experiments cover several classic ILP families and one non-binary integer dataset (NBI). Reported results indicate near-perfect feasibility rates and improvements in primal gap/integral, and the method can be used standalone or as a front-end to SCIP/Gurobi. Overall, the direction is interesting and relevant: pushing learning methods to reliably generate feasible ILP solutions is practically important. However, the experimental comparisons against recent learning-based baselines are limited, the evaluation on general integer variables is too narrow, and the paper omits crucial training curves and wall-clock analyses needed to judge compute cost and convergence behavior.

**Strengths:**

1. The paper provides a concrete demonstration that an RL policy can be trained end-to-end to produce feasible ILP solutions without solver repair, which is a nontrivial and interesting.

1. Unlike many learning-based works that focus only on binary variables, the method is designed for general integer domains and includes a non-binary benchmark.

**Weaknesses:**

1. The considered learning-based baselines are too limited. The paper mainly compares against PAS but does not include more recent learning approaches such as ConPS [1] and methods that explicitly generate feasible solutions such as [2] and [3]. Notably, [3] also trains in an unsupervised manner and reports high feasibility. Specifically, [3] adopts a gradient-based method, and the paper should discuss and compare the difference between RL-based and gradient-based methods.

1. The claim of handling general integer variables is under-evaluated. Only a single non-binary dataset (NBI) is used to support this point. More diverse general-integer tasks, or MIPLIB instances, are necessary to support this claim.

1. There are no training curves and no direct comparison of training compute with that of other learning baselines. Likewise, the paper should report test-time solving curves and wall-clock primal integral (PI) under identical hardware and thread settings.

1. The paper’s metric definitions and timing protocols are under-specified. In particular, PI depends on how time is discretized; if “timestep” refers to MDP steps for RL but wall-clock for non-RL baselines, the comparison is not sound.

[1] Huang et al., Contrastive Predict-and-Search for Mixed Integer Linear Programs. ICML 2024.

[2] Zeng et al., Effective Generation of Feasible Solutions for Integer Programming via Guided Diffusion. SIGKDD 2024.

[3] Geng et al., Differentiable Integer Linear Programming. ICLR 2025.

**Questions:**

1. Why choose Actor-Critic rather than more usually used algorithms such as SAC or PPO?

1. How exactly is PI computed in your experiments? Is “timestep” defined as MDP steps or wall-clock time? If it is MDP steps for RL but something else for baselines, the metric is not comparable. If it is wall-clock, please clarify the time interval.

1. In Algorithm 1, is the search routine used only during training to collect trajectories, or is the same search policy executed during testing as the deployed heuristic?

1. At test time, do you iterate the MDP for a fixed horizon, or do you use a termination rule? Please detail the stopping criteria, and how it interacts with solver calls when RL-SPH is used as a front-end.

---

> ### Author Response · Authors · 2025-11-21
>
> Thank you for providing constructive comments that help us more thoroughly validate our proposed method.
> We have made every effort to address all the concerns and questions you raised, and we would appreciate it if you could let us know if any further clarification is needed.
>
> # A.1 Comparing with end-to-end learning method
> > "The considered learning-based baselines are too limited. [...] the paper should discuss and compare the difference between RL-based and gradient-based methods"
>
> To the best of our knowledge, no learning-based *start primal heuristic* that converts an infeasible solution into a feasible one for ILP has been reported to achieve a 100\% feasibility rate (FR).
> Even when we broaden the scope to end-to-end learning primal heuristics (E2EPH), these methods still struggle to achieve 100\% FR without relying on external solvers.
> Since our focus is on finding feasible solutions, we considered it reasonable to compare our work with prior studies on E2EPH methods through the literature review presented in the related work section.
>
> For a more concrete comparison with recent E2EPH methods, the table below summarizes the overlap between the datasets used in our study and those employed in two recent top-tier conference papers [1,2].
> According to the results reported in [1], even when the proposed method is combined with SCIP’s CompleteSol heuristic, it still fails to obtain 100\% feasible solutions on CA.
> The average feasible ratios reported in [2] are 50.8\%, 97.1\%, and 99.4\% on SC, IS, and CA, respectively.
> Therefore, we consider classical heuristics such as Rounding and FP—both of which achieve higher FR on these datasets—to be stronger and more appropriate baselines for our evaluation.
>
> | | | IS | | CA | | SC |
> |:-----:|:--------------:|:--------------:|:--------------:|:--------------:|:--------------:|:--------------:|
> | Paper |   # variables  |  # constraints |   # variables  |  # constraints |   # variables  |  # constraints |
> |  [1]  | 1,500  | 6,396  | 1,500  | 786  | 2,000  | 1,000  |
> |  [2]  | 1,500  | 5,943  | 1,500  | 576  | 2,000  | 3,000  |
> |  Ours | 1,500  | 5,941  | 4,000  | 2,677  | 3,000  | 2,000  |
>
> As you pointed out, however, providing direct comparisons with other learning-based methods would further strengthen our claims.
> During the remaining discussion period, we will make every effort to conduct such direct comparisons with learning-based methods, should our current justification be considered insufficient.
> In Appendix G, we also discuss how RL-SPH differs from prior approaches such as E2EPH and LNS and how they could potentially be integrated with RL-SPH in future work.
>
> **Remark: Due to the leakage issue in OpenReview, further discussion with the reviewers has become impossible, leaving us uncertain about whether this rebuttal is sufficient. Thus, we summarize in Response A.8 the comparison with the latest E2EPH.**
>
> [1] "Effective generation of feasible solutions for integer programming via guided diffusion." KDD. 2024.
>
> [2] "Differentiable integer linear programming." ICLR. 2025.
>
> # A.2 Joint model on MIPLIB instances having integers
> > "The claim of handling general integer variables is under-evaluated"
>
> We updated our evaluation of the joint RL-SPH model, trained on IS+MVC (see Section 4.4), by including additional MIPLIB instances that have non-binary integers.
> The table below shows the statistics of the newly added instances.
> These instances differ from the training instances for the joint model in constraint type, density, size, and variable type.
>
> | Instance | Type | #variables | #binaries | #integers | #constraints | Density | BKS |
> |:---:|:---:|:---:|:---:|:---:|:---:|:---:|:---:|
> | gen-ip054 | General Linear | 30 | 0 | 30 | 27 | 65.70% | 6840.97 |
> | neos-3530903-gauja | Mixed Binary | 2,310 | 420 | 1,890 | 220 | 0.87% | 168 |
> - Statistics of instances from the official MIPLIB website. It also reports the objective value of the best known solution (BKS). To our knowledge, the website does not provide information on how the BKS was obtained and how much computation time was used.
>
> Nevertheless, RL-SPH obtains feasible solutions for these instances, as shown in the table below.
> In contrast, Rounding, the strongest baseline in terms of FR, fails to find a feasible solution for *neos-3530903-gauja*.
> Notably, *neos-3530903-gauja* contains a mixture of binary and integer variables, which differs substantially from the training instances (IS and MVC) that include only binary variables.
> This result demonstrates RL-SPH generalizes problem-solving capabilities to unseen instances, even when the variable type differs.
>
> | Test instance | Rounding | RL-SPH(IS+MVC) | BKS |
> |:---:|:---:|:---:|:---:|
> | gen-ip054 | 7148.68 | 7061.01 | 6840.97 |
> | neos-3530903-gauja | infeasible | 476 | 168 |
> - Comparison of RL-SPH's incumbent values against Rounding and BKS, where lower values indicate better performance. The time limit for Rounding and RL-SPH is set to 100 seconds.

---

> ### Author Response · Authors · 2025-11-21
>
> # A.3 Test-time solving curves
> > "paper should report test-time solving curves"
>
> We added test-time solving curves and training curves to the revised manuscript in Section 4.2 and Appendix J.7, respectively.
> We would also like to draw your attention to Figure 10 in Appendix J.6.1, which may be of particular interest.
>
> # A.4 Timestep defined as wall-clock time
> > "Is “timestep” defined as MDP steps or wall-clock time?"
>
> First of all, we sincerely apologize for the confusion caused while reading the paper.
> The timestep $t$ used in PI refers to the actual wall-clock time rather than the iteration count, and we should have stated this more clearly.
> Accordingly, we have revised Section 4.1.3 (Metric), and we appreciate your comment which allowed us to clarify this point.
> In addition, we have added a new metric, FT (the time required to obtain the first feasible solution), to our main experiment presented in Table 1.
>
> # A.5 Started with the basic Actor-Critic algorithm
> > "Why choose Actor-Critic rather than more usually used algorithms such as SAC or PPO?"
>
> Existing evidence shows that it is sufficiently effective for optimization problems [1].
> We believed that the key challenge lay in defining appropriate action spaces and reward structures for ILPs, whereas the selection of a more advanced RL algorithm was of secondary importance.
> Our plan was to explore alternative algorithms such as SAC or PPO only if AC showed insufficient performance or if specific issues arose that required a different approach.
> However, AC was able to train the model effectively.
> For this reason, we did not proceed with applying other algorithms.
>
> [1] Zong, Zefang, et al. "Mapdp: Cooperative multi-agent reinforcement learning to solve pickup and delivery problems.", AAAI, 2022.
>
> # A.6 Same search routine for training and testing
> > "is the search routine used only during training to collect trajectories?"
>
> Algorithm 1 is used as the same search routine for both training and testing.
> For clarity, we explicitly state in Section 3.3 that Algorithm 1 serves as the same search routine for both training and testing.
>
> # A.7 Stopping criteria
> > "Please detail the stopping criteria, and how it interacts with solver calls when RL-SPH is used as a front-end"
>
> The termination condition was set to a fixed wall-clock time predetermined according to the nature of each experiment, and we revised Section 4.1.2(Baselines) of the manuscript to make this clearer.
> When RL-SPH is used as a front-end, it determines which variables to fix by ensembling the collected feasible solutions, after which ILP solver fixes those variables using its built-in method before performing optimization, as described in Section 4.3.

---

> > ### Author Response · Authors · 2025-12-02
> >
> > # A.8 Comparison with SOTA E2EPH
> >
> > To further strengthen our claim, we also present empirical evidence in Appendix J.1 by experimentally comparing RL-SPH with recent E2EPH methods.
> > In accordance with the request of reviewer Hj6o, we conducted a comparison with DiffILO [4], the latest E2EPH method, which has been reported to generate better feasible solutions than other E2EPH methods [2,3].
> > In addition, following the request of reviewer pNKT, we also included the GNN-guided predict-and-search method (PAS) [1] in this evaluation.
> > In our experimental setup, DiffILO was trained on each dataset according to its original configuration.
> > Notably, its training time was approximately 34× longer than that of RL-SPH, indicating that DiffILO was given sufficient training time.
> > Our results show that only RL-SPH and Rounding consistently achieved a 100\% feasibility rate (FR) across all datasets.
> > In contrast, DiffILO achieved 100\% FR on IS, but it failed to find any feasible solution on CA and SC.
> > Further details are provided in Appendix J.1.
> >
> > Reference [2] does not appear to provide publicly available code, and due to time constraints, we were unable to include [3] in our experimental evaluation.
> > Nonetheless, given that [3] did not achieve 100\% feasibility even on datasets smaller than those used in our study, as shown in the table of Response-A.1, we expect that it would still struggle to reach 100\% feasibility on our datasets with numerous constraints.
> > We respectfully request the reviewers’ understanding that we have made every effort to address their requests within the limited time available.
> >
> >
> > [1] Han et al., A gnn-guided predict-and-search framework for mixed-integer linear programming. ICLR, 2023.
> >
> > [2] Huang et al., Contrastive Predict-and-Search for Mixed Integer Linear Programs. ICML 2024.
> >
> > [3] Zeng et al., Effective Generation of Feasible Solutions for Integer Programming via Guided Diffusion. SIGKDD 2024.
> >
> > [4] Geng et al., Differentiable Integer Linear Programming. ICLR 2025.

---

### Official Review · Reviewer_EHiR · 2025-10-30

**Soundness:** 3
**Presentation:** 2
**Contribution:** 3
**Rating:** 6
**Confidence:** 5

**Summary:**

This paper addresses the challenging problem of learning-based primal heuristics for integer linear programming (ILP), where existing end-to-end approaches often fail to produce feasible solutions independently. The authors propose RL-SPH, a reinforcement learning–based start primal heuristic that can generate feasible solutions—even for non-binary ILPs—with a 100% feasibility rate. Empirically, RL-SPH achieves significantly better performance than existing heuristics, demonstrating its practical promise for rapidly obtaining high-quality feasible solutions.

**Strengths:**

- The work tackles an important and underexplored problem: using learning-based methods to construct initial feasible solutions for general ILPs—a task known to be highly challenging but with substantial practical impact.
- The related work section is thorough and, to the best of my knowledge, covers nearly all recent relevant literature, which is commendable.
- The proposed method is conceptually simple yet highly effective. While the technical novelty may not be groundbreaking, the contribution lies in successfully navigating a difficult and sparsely explored direction, supported by key insights and theoretical justification.
- The experimental evaluation is comprehensive, including comparisons with classical heuristics, cross-problem generalization, and compatibility across different solvers, which collectively strengthen the empirical claims.
- The appendix is well-structured and contains valuable supplementary material and extended experiments. (That said, due to time constraints, I was unable to review it in full; a brief summary of appendix content in the main text would be helpful.)

**Weaknesses:**

- Several technical details are insufficiently explained, particularly in Section 3.2:
  + Figure 4 is unclear; it would help to explicitly illustrate the inputs and outputs of the actor and critic in each phase, and clarify which components constitute the Transformer’s sequence input.
  + The integration of structural information (e.g., from the constraint matrix) with variable features needs elaboration—how is this multi-dimensional structure encoded and concatenated?
  + It is ambiguous what exactly serves as the Transformer input: are variable nodes treated as a sequence (as in standard Graph Transformers)? Are phase indicators, objective coefficients, or other global features injected, and if so, in what form (e.g., as tokens, positional encodings, or side inputs)?
  + The ablation study shows the Transformer’s benefit, but the source of this advantage—e.g., long-range dependency modeling, attention over constraints, etc.—is not analyzed. Clarifying the key mechanism would greatly aid future work.
- The paper’s overall clarity and organization could be significantly improved. For instance, Section 3.3 appears to describe a component that logically precedes Figure 2, yet is placed afterward. The subdivision of Section 3.1 is overly granular, and Section 3.1.4 (training) is oddly positioned before Section 3.2. The text and figures also feel crowded, likely due to page limits; the authors should consider moving less critical details to the appendix to prioritize core content in the main body.
- The reward design in Section 3.1.3 is complex and central to the RL training, yet lacks motivation. A clear rationale—e.g., design principles, trade-offs considered, or inspiration from optimization theory—would help readers better appreciate the method’s novelty.
- The feasibility-aware search strategy in Section 3.3 appears somewhat heuristic. Given its empirical importance, the authors should clarify whether it draws from classical methods or is newly designed, what its computational cost is, whether it is a core contribution, and whether there is room for further optimization.

**Questions:**

- Could the authors clarify the technical details raised in the first weakness point, especially regarding the Transformer input construction and feature integration?
- The method achieves 100% feasibility even on non-binary ILPs, which is impressive given the vast search space. Were any special techniques employed to handle large or unbounded integer domains?
- The overall framework—iteratively setting and updating variable values—is relatively simple. In the authors’ view, what is the key enabler of its success: the reward design, the Transformer architecture, the feasibility-aware search strategy, or their combination?
- In Section 4.3, what happens if RL-SPH fails to find a feasible solution within 5 seconds? Is there a fallback mechanism, or is the run simply counted as a failure?
- The results on selected MIPLIB instances are compelling. Would it be possible to report performance on a broader set of instances to further strengthen the empirical claims?
- What do the authors see as the main bottleneck limiting scalability to very large-scale ILPs? Is it the Transformer architecture (e.g., quadratic attention), the RL training overhead, or something else?

**Overall, I consider this a high-quality contribution. While it may not introduce a paradigm-shifting algorithm, it demonstrates a viable and effective learning-based path in a notoriously difficult area—feasible solution generation for general ILPs—which I believe constitutes a meaningful advance. I will actively participate in the rebuttal discussion and look forward to a more comprehensive and objective evaluation based on the authors’ responses.**

---

> ### Author Response · Authors · 2025-11-21
>
> We are grateful for your positive assessment of our work’s contributions.
> Thank you also for the detailed comments regarding revisions to the paper.
> We have revised most of the relevant sections and will make every effort to incorporate the remainder during the discussion period.
>
> # A.1 Regarding paper revision
> ## A.1.1 Transformer input construction
> > "Could the authors clarify the technical details [...] regarding the Transformer input construction?"
>
> The phase indicator, objective value, and feasibility state serve as global features, and each of them is encoded as a token in our Transformer input.
> Each selected variable node can be viewed as a token.
> These global features and variable nodes are then fed into the Transformer as a sequence, as in standard Graph Transformers.
>
> Each variable has a vector of its objective coefficient and its coefficients across all constraints (i.e., the column of $\mathbf{A}$ mapped to that variable), which serves as its structural information.
> The structural information is projected to the same hidden size as the other features first and then concatenated with them.
> The final hidden feature is obtained by projecting the concatenated one to the predefined hidden size.
>
> To examine whether the proposed Transformer learns long-range variable dependencies, we conduct a case study showing attention weights between distant variables.
> As we believe that a figure-supported explanation is more effective, we respectfully ask you to refer to Appendix J.6.2 in the revised paper.
>
> We will revise Figure 4 and its description during the discussion period to address your comments and improve its clarity.
>
> ## A.1.2 Paper’s organization
> > "The paper’s overall clarity and organization could be significantly improved"
>
> We moved Section 3.1.4 (Training) to the end of Section 3.
> Since the actor–critic concept must be introduced before the proposed GNN architecture, we moved the description of the actor–critic, originally located in Section 3.1.4, to the preliminary section.
>
> As you pointed out, we also had concerns regarding the ordering of Section 3.3.
> Because our search strategy cannot be clearly explained without first introducing key RL components and GNN-related concepts, we decided to keep only the detailed content in this section.
> For the remainder, we relocated the conceptual overview to the preliminary section to help readers more easily understand Section 3.
>
> As we are able to use an additional page during the discussion period, we increased spacing between the text and figures.
> We found the RL-SPH+Gurobi experiment to be somewhat redundant, so we moved it to the appendix.
> Instead, we included test-time solving curves and a generalization experiment on MIPLIB instances in the main body, which we believe will be of greater interest to readers.
>
> We added a brief summary of the appendix at the end of the main body, as you suggested.
> We also added a table of contents at the beginning of the appendix to provide a clear overview of the appendix structure.
> If you have any suggestions on the paper revision, we would greatly appreciate your feedback.
>
> ## A.1.3 Clear rationale on reward design
> > "The reward design in Section 3.1.3 is complex and central to the RL training, yet lacks motivation"
>
> We added the full version of the below explanation to Appendix H.1, and rearranged the order of the conditions in Eq. 4 so that the priorities described below are more clearly reflected.
>
> As defined in Section 2.1, ILP minimizes an objective value while satisfying a set of constraints: linear constraints, integrality constraints, and variable bounds.
> When framing ILP within an RL system, the reward mechanism must be designed to reflect this definition.
> Thus, we derived the fundamental types of rewards: those related to objective values, linear constraints, and variable bounds.
> The reward for integrality constraints is omitted as they are handled by the action policy (see Section 3.1.1).
>
> According to the above definition, satisfying the set of constraints is a prerequisite for optimization.
> Therefore, rewards for the set of constraints should take high priority over the objective-value reward.
> It is intuitively reasonable that the more a constraint is violated, the greater the penalty should be.
> Likewise, a larger reward is appropriate as the objective value improves.
> From an optimization perspective, improving the feasibility of linear constraints while violating variable bounds is meaningless.
> For example, in an ILP where variables must take values of either 0 or 1, assigning a value such as –1 is invalid.
> Thus, satisfying the variable bounds has the highest priority among the variable bounds, linear constraints, and objective values.
> Additionally, to avoid the bound penalty being overly dominated by the constraint reward, we scaled it by the square root of the number of changeable variables.
> The maximum value of the objective coefficients scales the reward regarding the objective value.

---

> ### Author Response · Authors · 2025-11-21
>
> # A.2 Novelty of the feasibility-aware search strategy
> > "authors should clarify whether it draws from classical methods or is newly designed, what its computational cost is, whether it is a core contribution, and whether there is room for further optimization"
>
> The feasibility-aware search strategy was newly designed and not adapted from prior techniques.
> We are not aware of classical methods that resemble our feasibility-aware search strategy.
> Although the concept of exploring a neighborhood is borrowed from local search as described in the preliminary section, the variable selection policy itself was newly designed.
>
> The computational complexity of Algorithm 1 is determined by three components: variable selection, Transformer processing, and observation computation.
> For variable selection, multiplying the feasibility state vector $\tilde{\mathbf{f}}_t \in \mathbb{R}^{m \times 1}$ with the coefficient matrix $\tilde{\mathbf{A}} \in \mathbb{R}^{m \times n}$ requires $O(mn)$ operations.
> Processing $\tilde{n}^2 + 3$ input tokens with the Transformer incurs a cost of $O(\tilde{n}^2)$.
> For the observation update, only the changeable variables need to be recomputed, which results in $O(m \tilde{n})$.
> Therefore, the overall computational complexity of Algorithm 1 is $O(mn + \tilde{n}^2)$.
>
> We consider the feasibility-aware search strategy a key contribution, as it offers insight into how the search space can be reduced to enable more efficient learning for the RL agent.
> To better highlight this contribution, we revised the description of the search-strategy contribution in the introduction.
> We believe that the proposed search strategy represents only one possible way to reduce the search space, and that further optimization is certainly possible.
> For example, variable selection currently takes $O(mn)$, which becomes quadratic when $m = n$.
> Reducing this cost remains an interesting direction for future work.
>
>
> # A.3 Handling unbounded integers
> > "Were any special techniques employed to handle large or unbounded integer domains?"
>
> When we began this work, we anticipated that directly regressing variable values in ILP with unbounded integer domains would be extremely challenging.
> Because the variable ranges are unbounded, it is practically impossible for a regression model to predict *exact* integer values without any error.
> Such inaccurate predictions would inevitably require additional post-processing such as rounding, which introduces further errors.
> Even if a model were hypothetically able to output perfect integers without any post-processing, the resulting combination of predicted variable values could still be infeasible.
> Obtaining feasible solutions remained challenging, as we had already observed in prior studies even on binary ILP.
>
> This led us to conclude that predicting all variable values directly in unbounded integer domains in one shot would be an extremely difficult task.
> For this reason, we sought to simplify the problem by reformulating it so that the agent incrementally decides whether to increase or decrease each variable over multiple steps.
> We considered this iterative decision-making approach to be both more realistic and more effective.
>
> However, even with this simplification, designing the observation space for the RL formulation posed a significant challenge.
> It is essential to preserve the numerical information contained in ILP while applying appropriate scaling to ensure that the ML model could learn effectively.
> Typical ILP instances contain unbounded values not only for variables but also for coefficients and RHS terms, which magnifies the difficulty.
> To address this, we investigated methods for scaling LP without altering the underlying problem.
> As described in Section 3.2, we applied equilibration scaling when feeding ILP instances into the ML model.
> We also surveyed prior ML work on numerical tasks and incorporated Periodic Embedding (PE), which are known to be effective for numerical features.
> We conducted an ablation study on PE, but since the performance differences were not substantial, we did not include the results in the paper.

---

> ### Author Response · Authors · 2025-11-21
>
> # A.4 Trinity as the key enabler
> > "what is the key enabler of its success: the reward design, the Transformer architecture, the feasibility-aware search strategy, or their combination?"
>
> Since none of the three components can be removed without compromising the performance reported in the paper, the key enabler is the combination of all three.
> The ablation studies in Section 4.5 show that RL-SPH struggles to find a feasible solution quickly without our search strategy and RL-SPH achieves the best results with all components of our GNN, supporting this claim.
>
> These three components interact closely with one another, and each component increases the overall complexity of the system.
> - **The reward design**: A poorly designed reward function may introduce loopholes that the agent exploits undesirably. For example, the agent may remain stationary to avoid penalties. Additionally, the objective function may outweigh the constraints, leading to violations. To ensure appropriate behavior, the reward function must be carefully crafted (see Appendix H.1), and the GNN must be designed to be context-aware (see RC in Table 6).
> - **The feasibility-aware search strategy**: Even with a well-designed reward function and GNNs, introducing irrelevant inputs can slow down training convergence. Even if training succeeds, noisy inputs can degrade solving performance. Consequently, narrowing the search space facilitates more efficient learning and enhances solving effectiveness (see Appendix K.4 and Table 5).
> - **The Transformer architecture**: Furthermore, after the variable selection step, the full bipartite graph is no longer available to the model. It means that some bridge nodes are lost and certain reachability paths disappear, as shown in blank cells in Figure 11(a). Consequently, an MPNN-based GNN may struggle to learn effective representations for such isolated variable nodes. Thus, the GNN must be designed to effectively capture long-range dependencies.
>
> Given their tight interaction, the combination of these three components is the key enabler of RL-SPH's success.
> At the same time, we acknowledge that better designs for each individual component may certainly exist and represent promising avenues for future research.
>
> # A.5 Failure handling in RL-SPH
> > "what happens if RL-SPH fails to find a feasible solution within 5 seconds? Is there a fallback mechanism, or is the run simply counted as a failure?"
>
> In Section 4.3, if RL-SPH fails to find a feasible solution within 5 seconds, the run would be recorded as a failure.
>
> # A.6 Joint model on MIPLIB instances having integers
> > "The results on selected MIPLIB instances are compelling. Would it be possible to report performance on a broader set of instances to further strengthen the empirical claims"
>
> We demonstrate that RL-SPH(IS+MVC) generalizes to unseen instances (*gen-ip054* and *neos-3530903-gauja*) containing non-binary integer variables.
> To reduce the redundant explanation, we discuss MIPLIB-related topics in A2 of our response to Reviewer Hj6o.
> We kindly encourage you to refer to that section for further details.
>
> # A.7 Potential bottlenecks at extreme scales
> > "What do the authors see as the main bottleneck limiting scalability to very large-scale ILPs?"
>
> While our search strategy alleviates the scalability issue by passing only $\tilde{n}(=2 \left\lceil\log_2n\right\rceil)$ inputs to the Transformer, we agree that the quadratic complexity of the Transformer’s attention mechanism could become a bottleneck for extremely large-scale problems.
> For such cases, it would be beneficial to consider attention architectures specifically designed for lower computational complexity, such as RetNet [1] or other variants.
>
> Even if the quadratic attention in the transformer is resolved, the variable selection mentioned in A2 may become a new bottleneck.
> Addressing this issue will likewise be an important direction for future work.
>
> Furthermore, if the number of constraints becomes very large, projecting the structural information into the hidden size may incur substantial computational cost.
> This issue could be addressed by explicitly introducing constraint nodes as part of the Transformer input tokens, rather than compressing all structural information into a single projected vector.
> We conducted preliminary experiments to explore this direction and observed promising performance, although more thorough evaluation is certainly required.
>
> [1] Sun, Yutao, et al. "Retentive network: A successor to transformer for large language models." arXiv preprint arXiv:2307.08621 (2023).

---

> > ### Comment · Reviewer_EHiR · 2025-11-26
> >
> > Thank you for your response. My concerns were largely about technical and presentation details, and the authors have addressed them well. For clarity, I recommend incorporating some important clarifications into the paper (e.g., updating Figure 4), following some of my previous sugesstions.
> >
> > Separately, to better position the contribution, I suggest explicitly discussing how this work relates to conceptually relevant prior approaches. While E2EPH and LNS are not direct baselines, placing them (and the works [2, 3] cited by Reviewer Hj6o, which appear more closely related) in a unified background—clarifying distinctions and potental connections—would help readers grasp the novelty and contributions.

---

> > > ### Author Response · Authors · 2025-11-26
> > >
> > > Thank you for your constructive suggestions. As promised, we have clarified Figure 4 during the discussion period and updated the corresponding explanation in Section 3.2.
> > >
> > > We also consolidated our rebuttal responses regarding the positioning of RL-SPH into Appendix G. In this appendix, we explain how RL-SPH conceptually differs from E2EPH and LNS, and how these approaches could potentially be integrated in future work.
> > >
> > > We hope these revisions help readers more clearly understand the novelty and contributions of our work. We would be grateful for any additional feedback or suggestions you may have.

---

> ### Comment · Reviewer_EHiR · 2025-11-27
>
> Thank you for your timely and thoughtful response, as well as the supplementary material.  I find the newly added Appendix G highly relevant and informative.  To further clarify the novelty and scope of this work, I suggest the authors more explicitly delineate the fundamental differences between their approach and key prior methods, with emphasis on both the methodological distinctions and the increased challenge of the problem formulation studied here (e.g., why focus on feasibility here, rather than the feasibility + optimization mode in E2EPH). Such clarification would greatly help address concerns raised by other reviewers.
>
> In addition, as the authors themselves acknowledge, including comparisons with the latest learning-based E2EPH methods would substantially strengthen the empirical evaluation and enhance the paper’s overall impact.
>
> Taking the rebuttal and ongoing discussion into account, I am currently inclined to retain my original positive rating, accompanied by a high confidence score. I look forward to seeing these points addressed in the revision and remain willing to actively participate in discussions with the authors, fellow reviewers and the Area Chair to ensure a thorough and balanced assessment.

---

> > ### Author Response · Authors · 2025-11-27
> >
> > Thank you for your detailed feedback, which has helped improve the quality of our paper.
> >
> > As suggested, we highlighted the distinctions between RL-SPH and prior approaches in Appendix G to further clarify the novelty and scope of our work.
> >
> > We appreciate your interest in our research and will continue to engage fully in the discussion for the remainder of the review process.

---

> ### Author Response · Authors · 2025-12-02
>
> To strengthen our claims with the empirical evaluation and enhance the paper’s overall impact, we have added comparisons with the latest learning-based E2EPH method in Appendix J.1.
> Accordingly, we have further revised Appendix G (Positioning relative to prior work).
>
> In addition, we believe that several of our rebuttal responses contain information that would be useful to readers, so we have included the following items in the discussion section of the Appendix.
> - A.2 Novelty of the feasibility-aware search strategy
> - A.4 Trinity as the key enabler
> - A.7 Potential bottlenecks at extreme scales
>
> We have also clarified the computational cost of Algorithm 1—previously explained in our rebuttal—in Appendix B.
> Finally, due to space limitations, we could not include the content of rebuttal response A.5 (Failure handling in RL-SPH) in the main body, so we added the corresponding explanation to Appendix J.3, where the same experiment is presented.
>
> We regret that the OpenReview leakage issue prevents us from continuing the discussion.
> We deeply appreciate your strong interest in our research and the constructive feedback you offered during the discussion period, which has significantly contributed to strengthening our manuscript.

---

### Official Review · Reviewer_m7aJ · 2025-10-31

**Soundness:** 2
**Presentation:** 3
**Contribution:** 2
**Rating:** 4
**Confidence:** 3

**Summary:**

The paper introduces RL-SPH, a reinforcement-learning-based start primal heuristic that can independently produce feasible solutions for ILPs (including non-binary integers). Key components:
1) Bipartite graph encoding of the ILP; a transformer-based GNN outputs per-variable actions {−1, 0, +1}.
2) Two-phase reward: phase 1 maximises feasibility improvement; phase 2 maximises objective improvement after the first feasible solution.
3) Feasibility-aware variable selection: only variables appearing in currently violated constraints are made “changeable”, reducing the action space.
Trained with Actor-Critic on five NP-hard benchmarks, RL-SPH attains 100 % feasibility rate, 44 × smaller primal gap and 2.3 × lower primal integral than built-in start heuristics of SCIP, and generalises to larger, denser and out-of-distribution instances as well as to MIPLIB.

**Strengths:**

1) Guarantees feasibility even from random initial solutions, overcoming a major limitation of previous ML-based heuristics.
2) Creative use of transformer-GNN plus phase-separated actor-critic to capture long-range variable-constraint interactions.
3) Strong empirical performance across five problem classes and robust generalisation to size/density/distribution shifts and MIPLIB instances.

**Weaknesses:**

1) Action magnitude is restricted to ±1; preliminary results show larger moves improve objective but are not pursued, leaving optimality on the table.
2) Training requires thousands of episodes on 64 parallel instances; total GPU-hours and carbon footprint are not reported.
3) Only compares against SCIP’s internal heuristics; no comparison with recent large-neighbourhood-search or diffusion-based heuristics.
4) The polynomial-time guarantee relies on the assumption that the agent always improves feasibility; in practice exploration may stall—no discussion of timeout handling.

**Questions:**

1. What is the computational cost (GPU-hours and energy) of training RL-SPH on the largest benchmark, and how does it compare with collecting near-optimal labels for supervised methods?
2. Could dynamically increasing the action magnitude (e.g., ±5) when the agent is stuck improve final objective value without harming feasibility?
3. How would RL-SPH perform as a standalone solver on very large instances (10^6 variables) where LP relaxation is expensive, and could problem decomposition be integrated?

---

> ### Author Response · Authors · 2025-11-21
>
> Thank you for your insightful comments and for providing thought-provoking feedback that fosters constructive discussion.
> We respond to your concerns in detail below.
> # A.1 Action magnitude
> > "Action magnitude is restricted to ±1; preliminary results show larger moves improve objective but are not pursued, leaving optimality on the table"
>
> As you pointed out, we do not place emphasis on optimality, since our focus lies on feasibility rather than optimality, as discussed in Section 5.
> Obtaining feasible solutions remained challenging, as we had already observed in many prior studies even on binary ILP.
> In particular, even recent studies [1,2] published in top-tier conferences still fail to achieve a 100% feasibility rate (FR), as detailed in A1 of our response to Reviewer Hj6o.
> Although our primary focus is feasibility, we note that RL-SPH also exhibits strong solution quality: even with an action magnitude of ±1, it outperforms four classical start primal heuristics and the SOTA E2EPH [2].
> As Reviewer EHiR acknowledges, given this notoriously challenging nature of the feasibility issue, our study necessarily focuses on addressing feasibility first, leaving the investigation of optimality to future work.
> Given that our study focuses on feasibility and an action magnitude of ±1 already yields strong performance, we believe that exploring larger action magnitudes would not affect our main conclusions.
>
> [1] "Effective generation of feasible solutions for integer programming via guided diffusion." KDD. 2024.
>
> [2] "Differentiable integer linear programming." ICLR. 2025.
> # A.2 Computational cost
> > "Training requires thousands of episodes on 64 parallel instances; total GPU-hours and carbon footprint are not reported. What is the computational cost (GPU-hours and energy) of training RL-SPH on the largest benchmark, and how does it compare with collecting near-optimal labels for supervised methods?"
>
> Although we train on 64 parallel instances, each instance requires only $2 \left\lceil\log_2n\right\rceil+3$ input tokens for training (see Section 3.2), resulting in a relatively low training cost.
> As reported in Appendix I.3, the training times for the datasets were about 31, 32, 26, 83, and 23 minutes for IS, CA, SC, MVC, and NBI, respectively.
> RL-SPH was roughly 8.7× faster than a representative supervised learning method [1] and 34× faster than an unsupervised method [4] in terms of the training time, as shown in the below table.
> This substantial reduction in training time underscores the practicality of RL-SPH.
>
> As explained in Appendix K.3, near-optimal solutions are obtained by running Gurobi with a time limit of 60 minutes for a single instance [1,2].
> Considering that training requires on 400 instances in [1], this constitutes a substantial computational cost.
> This labeling process becomes even more computationally expensive for larger instances, limiting the practicality for NP-hard problems [3].
> Thus, RL-SPH incurs lower training cost than supervised learning methods, as it requires no labeling process and less training time.
>
> Given that RL-SPH trains in 39 minutes on average and is 34× faster than the SOTA E2EPH [4], we hope that the concern regarding its training cost can be reconsidered.
>
> Since we have not yet encountered any studies in the ML4CO field that report the carbon footprint, could you introduce any works that we could refer to?
>
> | Dataset | IS | CA |
> |:---:|:---:|:---:|
> | PAS | 209 | 343 |
> | RL-SPH | 31 | 32 |
> | Speedup |  **6.7×**  |  **10.7×**  |
>
> | Dataset | IS | CA | SC |
> |:---:|:---:|:---:|:---:|
> | DiffILO | 1,002 | 1,177 | 850 |
> | RL-SPH | 31 | 32 | 26 |
> | Speedup |  **32.3×**  |  **36.8×**  |  **32.7×**  |
>
> [1] "A GNN-Guided Predict-and-Search Framework for Mixed-Integer Linear Programming." ICLR. 2023.
>
> [2] "Contrastive predict-and-search for mixed integer linear programs." ICML. 2024.
>
> [3] "Scalable Primal Heuristics Using Graph Neural Networks for Combinatorial Optimization." Journal of Artificial Intelligence Research, 2024.
>
> [4] "Differentiable integer linear programming." ICLR. 2025.
>
>
> # A.3 Comparison with other heuristics
> > "Only compares against SCIP’s internal heuristics; no comparison with recent large-neighbourhood-search or diffusion-based heuristics"
>
> As large neighbourhood search (LNS) requires an initial feasible solution in order to operate [1], its scope within the solution process differs from that of start primal heuristics.
> For this reason, we respectfully note that LNS does not serve as a direct baseline for comparison with our method.
> In Appendix G, we further discuss how RL-SPH differs from prior approaches such as LNS and how they could potentially be integrated with RL-SPH in future work.
>
> Since we are not familiar with diffusion-based heuristics, would you be able to share any references that could help us understand more concretely what you mean by this term?
>
> [1] Primal heuristics for mixed integer programs. PhD thesis, Zuse Institute Berlin (ZIB), 2006.

---

> ### Author Response · Authors · 2025-11-21
>
> # A.4 Polynomial-time guarantee
> > "The polynomial-time guarantee relies on the assumption that the agent always improves feasibility; in practice exploration may stall—no discussion of timeout handling"
>
> We appreciate the reviewer’s insightful observation.
> It is correct that our polynomial-time guarantee was established under the assumption that the agent consistently improves feasibility (or equivalently, receives non-negative rewards for linear constraints and bound variables).
> We agree that this assumption does not necessarily hold in every practical circumstance.
> Therefore, we considered this assumption somewhat restrictive and removed it, introducing a new condition instead.
> To address this, we have:
> - Added the condition $\mathcal{T} \le n$ explicitly to Proposition 2,
> - Provided a detailed justification for this condition in the proof, and
> - Clearly stated in the introduction that the polynomial-time feasibility guarantee holds under this explicit condition.
>
> When imposing the condition $\mathcal{T} \le n$, the assumption that the agent always improves feasibility is no longer enforced; in other words, our revised proposition allows for stalls in feasibility improvement.
> Since $\mathcal{T}$ denotes the number of search steps needed to reach the first feasible solution, this condition is more relaxed and permits such stalls compared to our previous assumption.
>
> These revisions clarify the scope of our guarantee and prevent potential misinterpretation of the underlying assumptions.
>
> Regarding timeout handling: the search procedure terminates only when the predefined wall-clock time limit is reached.
> In all of our experiments, RL-SPH successfully found feasible solutions well within this limit.
> If, however, the agent fails to find a feasible solution before the time limit, it would be treated as a failure, consistent with standard practice followed by other start primal heuristics.
> We have clarified this behavior in Section 4.1.2 of the revised text to avoid ambiguity.
>
> # A.5 Improving the incumbent value without harming feasibility
> > "Could dynamically increasing the action magnitude (e.g., ±5) when the agent is stuck improve final objective value without harming feasibility?"
>
> Given that we suggested dynamically increasing the action magnitude as one of the promising ways to improve optimality, improving the final objective value can be interpreted as updating the incumbent objective value.
> In RL-SPH, the incumbent objective value is available only in $phase 2$.
> In $phase 2$, the policy is designed to roll back any move that worsens feasibility or fails to produce a better feasible solution (see Algorithm 1).
> Therefore, increasing the action magnitude when the agent becomes stuck does not lead to any deterioration in feasibility in RL-SPH.
>
> # A.6 Handling extremely large scale ILP
> > "How would RL-SPH perform as a standalone solver on very large instances where LP relaxation is expensive, and could problem decomposition be integrated?"
>
> If the LP relaxation is expensive for extremely large-scale problems, random initialization for RL-SPH can be considered as one possible approach.
> Our main experimental results demonstrate that RL-SPH with random initialization is able to begin solution exploration at an early stage, as explained in Section 4.2.
> Moreover, these findings motivate the development of an initialization strategy that is more refined than random assignment yet faster than LP relaxation.
> Problem decomposition, as you mentioned, is also a promising method for handling problems of extremely large scale, and further investigation will be necessary by drawing on the insights from [1] and [2].
>
> [1] "GNN\&GBDT-guided fast optimizing framework for large-scale integer programming." ICML, 2023.
>
> [2] "Light-MILPopt: Solving large-scale mixed integer linear programs with lightweight optimizer and small-scale training dataset." ICLR. 2024.

---

### Official Review · Reviewer_pNKT · 2025-11-02

**Soundness:** 3
**Presentation:** 2
**Contribution:** 2
**Rating:** 6
**Confidence:** 3

**Summary:**

This paper proposes RL-SPH, a novel end-to-end framework for learning to generate feasible and high-quality solutions for ILPs. RL-SPH can operate independently, though it optionally allows LP-relaxation initialization. The method introduces a two-phase reward system that explicitly separates feasibility recovery (Phase 1) and objective optimization (Phase 2). A key component of the algorithm is its feasibility-aware variable selection, where in each step the agent selects a subset of variables and decides whether to increment, decrement, or keep them unchanged. The selection heuristic prioritizes variables that contribute most to constraint violations, effectively narrowing the search space and improving learning stability. Extensive experiments on five benchmark datasets show that RL-SPH achieves 100\% feasibility rate, substantially reduces Primal Gap and Primal Integral.

**Strengths:**

The design of the reward is novel and optimization-inspired, and the two-phase reward system (feasibility → optimization) is one of the paper’s main contributions. It provides dense and directional feedback in regions where classical RL methods often suffer from sparse or unstable signals. It provides dense and directional feedback in regions where classical RL approaches suffer from sparse or misleading signals.

The “feasibility-aware variable selection” (Algorithm 3) is also a well-motivated component. By allowing the agent to modify only a small subset of variables (≈ 2 log n) that appear most frequently in violated constraints, the search remains interpretable and computationally manageable. This strategy reduces unnecessary exploration and empirically accelerates the recovery of feasible states.

Another strength is that RL-SPH can start from either LP-relaxation or pure random initialization, with the latter achieving nearly identical performance. This shows that the model learns feasibility patterns directly rather than relying on solver-provided starting points, which makes the approach more self-contained and practical.

Finally, the experimental evaluation is broad and convincing. The paper tests on five benchmark datasets, including a non-binary integer (NBI) case, demonstrating that the approach generalizes beyond 0–1 ILPs. RL-SPH consistently attains full feasibility and yields clear improvements in Primal Gap and Primal Integral compared to established heuristics.

**Weaknesses:**

The reward formulation involves several manually chosen hyperparameters (e.g., $\alpha$, $p$, $q$, etc). While the method performs well, it is unclear how sensitive the results are to these hyperparameters. A small sensitivity analysis would help confirm whether the learning dynamics are robust or heavily dependent on manual tuning.

The baselines used (Feasibility Pump, Diving, Rounding, PAS) are all classical heuristics. However, several recent learning-based feasibility or primal heuristics (e.g., GNN-guided predict-and-search methods or learning-to-repair approaches) could provide a fairer context for comparison. Including at least one recent end-to-end learning method, particularly an RL-based method, would better position RL-SPH within current literature.

Although ablation studies are included, they focus primarily on architectural components. The paper would benefit from a clearer quantitative analysis of the two-phase reward system (e.g., measuring feasibility rate after Phase 1 alone, or visualizing constraint violation) over time. This would help validate the intended role of each reward component and make the learning process more interpretable.

While the paper reports Primal Gap and Primal Integral metrics, it does not discuss actual wall-clock runtime. These metrics only indirectly reflect temporal progress and are normalized by iteration counts, not by real elapsed time. As a result, the practical efficiency of RL-SPH relative to classical heuristics or solver-based methods remains unclear. Reporting average runtime per instance would make the comparison more fair. This omission significantly weakens the empirical evaluation, as efficiency is a core claim of the paper.

**Questions:**

1. The −100 penalty effectively discourages idle behavior ($x_{t+1} = x_t$), but it cannot prevent oscillations between distinct but equivalent states (loops). Did the authors observe such cyclic behavior during Phase 1, and if so, how was it mitigated? Would adding a short-term memory or visited-state penalty help stabilize the feasibility recovery process?
2. The paper reports PG/PI/FR but no wall-clock runtime. Could the authors provide the average runtime per instance? This would make comparisons fairer.
3. The paper claims that “to the best of our knowledge, RL-SPH is the first end-to-end primal heuristic that explicitly learns feasibility for ILPs with a theoretical guarantee.” However, Tang, Khalil & Drgoňa [1] have already proposed a learning-based optimization framework with explicit feasibility guarantees for MINLPs, which subsumes ILPs as a special case. Could the authors clarify how RL-SPH differs conceptually or theoretically from this prior work, and in what sense it constitutes the “first” method with feasibility guarantees?
4. The paper shows that RL-SPH(Random) performs comparably to RL-SPH(LP) for 0–1 problems, but no corresponding experiment is reported for the non-binary integer (NBI) dataset. Was random initialization more difficult or unstable in this setting (e.g., due to unbounded variable ranges or larger search space)? Providing results or discussion for the NBI case would clarify whether the Random Initialization generalizes beyond binary ILPs.

[1] Tang, B., Khalil, E. B., & Drgoňa, J. (2024). Learning to Optimize for Mixed-Integer Non-linear Programming with Feasibility Guarantees. arXiv preprint arXiv:2410.11061.

---

> ### Author Response · Authors · 2025-11-21
>
> We thank you for your constructive review and for your interest in our work.
> Your valuable comments prompted additional analyses and discussions, which we believe will substantially enrich our paper.
> # A.1 Hyperparameter analysis
> > "A small sensitivity analysis would help confirm whether the learning dynamics are robust or heavily dependent on manual tuning."
>
> We conducted analyses on the hyperparameters you pointed out: $\alpha$, $p$, and $q$.
> The experiments below show that there is room for achieving even better performance with dataset-specific tuning.
> We emphasize that RL-SPH outperforms classical start primal heuristics and E2EPH methods even without dataset-specific tuning.
> We aimed to provide guidelines for choosing suitable hyperparameter values.
>
> ## A1.1 Effect of the toward-optimal bias
> The table below show the performance of RL-SPH under different values of the toward-optimal bias $\alpha$.
> With a toward-optimal bias (i.e., $\alpha>1$), RL-SPH achieves the better PG, PI, and \#win than the value obtained with $\alpha = 1$.
> In contrast, RL-SPH without such bias (i.e., $\alpha = 1$) consistently attains the lowest \#win across all datasets.
>
> Increasing the bias from $\alpha=2$ to $\alpha=5$ does not necessarily yield further improvements.
> Notably, in SC, RL-SPH($\alpha=5$) achieves a PG of 2.52\%, which is even higher than the value obtained with $\alpha=1$.
> Given that the model also exhibits a noticeably high value of \#viol\_2 in SC, we conjecture that the model becomes overly focused on pursuing a lower objective value, which in turn leads to more aggressive attempts that may even violate constraints.
> Therefore, it is advisable to choose $\alpha$ to be greater than 1 but not excessively large.
> | Dataset | $\alpha$ | FR (%)↑ | PG (%)↓ | PI↓ | #win↑ | #viol_2 |
> |:---:|:---:|:---:|:---:|:---:|:---:|:---:|
> | IS | 1 | 100 | 3.41±2.10 | 7.2±1.0 | 9 | 1.0±0.1 |
> |  | 2 | 100 | 0.94±1.29 | 3.4±0.6 | 45 | 0.3±0.1 |
> |  | 5 | 100 | **0.85±1.20** | **3.3±0.6** | **53** | 0.4±0.1 |
> | CA | 1 | 100 | 3.22±3.00 | 12.5±1.3 | 17 | 2.2±0.5 |
> |  | 2 | 100 | **0.88±1.64** | **11.8±1.1** | **57** | 2.1±0.4 |
> |  | 5 | 100 | 2.49±2.58 | 12.4±1.2 | 26 | 3.1±0.5 |
> | SC | 1 | 100 | 2.39±2.13 | 98.2±6.8 | 32 | 4.0±4.7 |
> |  | 2 | 100 | **1.74±2.22** | 98.4±7.2 | **50** | 4.7±6.3 |
> |  | 5 | 100 | 2.52±2.41 | **97.8±7.2** | 33 | 11.2±17.9 |
> | MVC | 1 | 100 | 14.18±0.90 | 8.8±0.4 | 0 | 1.7±0.2 |
> |  | 2 | 100 | **0.43±0.69** | **4.6±0.3** | **60** | 0.8±0.1 |
> |  | 5 | 100 | 0.67±0.83 | 4.7±0.4 | 41 | 1.1±0.1 |
> | NBI | 1 | 100 | 0.24±0.11 | 10.9±0.6 | 1 | 1.4±0.3 |
> |  | 2 | 100 | 0.06±0.07 | **10.7±0.6** | 29 | 2.7±0.6 |
> |  | 5 | 100 | **0.01±0.03** | 10.8±0.7 | **71** | 4.6±0.7 |
> - \#viol\_2 is the sum of variable bound violations and linear constraint violations in $phase2$.
>
> ## A1.2 Effect of the neighbor variable ratio
> We varied only the ratio of neighbor variables and fixed the number of input variables (i.e., $ \tilde{n}=p+q$).
> The table below presents the performance under different ratios of neighbor variables.
> Regardless of the ratios, RL-SPH finds a feasible solution in all settings.
> In IS and MVC, higher neighbor variable ratios enable RL-SPH to reach higher-quality feasible solutions quickly, suggesting that IS and MVC exhibit stronger interactions among neighbor variables than the others.
> | Dataset | Neighbor variable ratio | FR (%)↑ | PG (%)↓ | PI↓ | #win↑ |
> |:---:|:---:|:---:|:---:|:---:|:---:|
> | IS | 0.3 | 100 | 8.22±1.70 | 6.9±0.8 | 0 |
> |  | 0.5 | 100 | 5.86±1.94 | 5.7±0.9 | 0 |
> |  | 0.7 | 100 | **0.00±0.00** | **2.8±0.2** | **100** |
> | CA | 0.3 | 100 | **0.99±2.08** | **11.9±1.2** | **60** |
> |  | 0.5 | 100 | 1.57±2.11 | 12.1±1.2 | 40 |
> |  | 0.7 | 100 | 8.84±3.38 | 15.0±1.6 | 0 |
> | SC | 0.3 | 100 | 2.15±2.24 | 97.6±6.7 | 39 |
> |  | 0.5 | 100 | **1.59±2.10** | 98.2±7.4 | **52** |
> |  | 0.7 | 100 | 2.55±2.37 | **97.2±7.4** | 31 |
> | MVC | 0.3 | 100 | 7.49±0.94 | 7.8±0.4 | 0 |
> |  | 0.5 | 100 | 6.90±0.97 | 7.6±0.4 | 0 |
> |  | 0.7 | 100 | **0.00±0.00** | **4.8±0.2** | **100** |
> | NBI | 0.3 | 100 | **0.03±0.04** | 10.9±0.7 | **59** |
> |  | 0.5 | 100 | 0.05±0.06 | **10.7±0.6** | 32 |
> |  | 0.7 | 100 | 0.09±0.06 | 10.9±0.8 | 10 |
>
> As shown in the table below, IS and MVC have high variable degrees compared to their constraint degrees (i.e., a higher degree ratio).
> Consequently, when the agent modifies the value of a seed variable to improve feasibility, it must also account for the many constraints involving that variable, along with the other variables that appear in those constraints.
> Thus, for datasets with a large degree ratio, a larger neighbor variable ratio is preferable, as it allows the agent to capture interactions among more neighbors.
>
> | Dataset | IS | CA | SC | MVC | NBI |
> |:---:|:---:|:---:|:---:|:---:|:---:|
> | Degree of constraints $d_c$ | 2.00 | 8.31 | 150.00 | 2.00 | 198.98 |
> | Degree of variables $d_v$ | 7.94 | 5.56 | 100.00 | 7.96 | 198.98 |
> | Degree ratio $d_v / d_c$ | 3.96 | 0.67 | 0.67 | 3.98 | 1.00 |

---

> ### Author Response · Authors · 2025-11-21
>
> # A.2 Comparing with end-to-end learning method
> > "Several recent learning-based [...] could provide a fairer context for comparison"
>
> To reduce the redundant explanation, we discuss baseline-related topics in A1 of our response to Reviewer Hj6o.
> In Appendix G, we discuss how RL-SPH differs from prior approaches and how they could be integrated with RL-SPH.
> We kindly encourage you to refer to those section for further details.
>
> # A.3 Behavior of two-phase reward system
> > "This would help validate the intended role of each reward component"
>
> As we believe that an explanation with figure would be more effective, we kindly ask you to refer to Appendix J.6.1.
>
> # A.4 Regarding wall-clock time
> > "While the paper reports PG and PI, it does not discuss actual wall-clock runtime"
>
> We sincerely apologize for the confusion.
> The timestep $t$ used in PI refers to the wall-clock time.
> We revised Section 4.1.3, and we appreciate your comment, which allowed us to clarify this point.
> We added a new metric, FT (the time required to obtain the first feasible solution), in Table 1.
>
> # A.5 No loop in phase1
> > "Did the authors observe such cyclic behavior during Phase 1"
>
> Let us define equivalent states and loops as follows: we define equivalent states as solutions that differ in their assignments but have the same quality (i.e., identical objective values and violation degrees), and we define a loop as a situation in which the agent repeatedly revisits such equivalent states without making progress in the quality.
>
> If solutions have the same objective value and violation degree (i.e., equivalent states), any transition between them in $phase1$ yields a reward of zero.
> Once a loop occurs, the agent would receive zero reward in $phase 1$ repeatedly, which implies that it is no longer able to find a state better than the current one.
> Transitioning from $phase1$ to $phase 2$ requires converting an infeasible solution to a feasible one, which clearly indicates a transition to a better state.
> Thus, if a loop occurs in $phase 1$, the agent must fail to find any feasible solution, which contradicts our empirical results showing a 100\% FR.
> Therefore, we conclude that loops did not occur in our experiments.
>
> We attribute the absence of loops to our well-designed reward system and search strategy.
> As long as feasibility improves, the RL agent receives a non-negative reward for constraints and variable bounds, and an agent trained to maximize reward continues to improve feasibility.
> As shown in Figure 9, the total violation decreases over time, and this empirical evidence supports our claim.
> Moreover, since our search strategy samples variables probabilistically, the GNN is likely to receive different inputs even for equivalent states.
> We conjecture that such randomness would encourage the agent to escape loops.
>
> Although we did not observe loops, the approach you suggested would be quite interesting.
> Incorporating the short-term memory could guide the agent toward finding the better feasible solution.

---

> ### Author Response · Authors · 2025-11-21
>
> # A.6 Comparison of feasibility guarantees
> > "In what sense it constitutes the “first” method with feasibility guarantees?"
>
> We appreciate you bringing this interesting work [1] to our attention.
> It appears that we unintentionally omitted it during our literature review.
>
> Due to limited time, we were unable to fully grasp the theoretical foundations of [1], so we instead provide a conceptual comparison in terms of feasibility guarantees.
> As far as we understand, their theorem establishes feasibility by relating constraint violations to gradient convergence.
> Specifically, repeated gradient descent leads the gradient to converge to zero, which in turn implies zero violation.
> Their loss function is the sum of the loss for the objective function and the loss for constraint violations.
> However, the proof in [1] does not take the objective function into account.
> Moreover, considering that [1] introduces a penalty hyperparameter to control the trade-off between the two, it appears that no explicit priority is enforced between them.
> In addition, in their ICLR 2025 submission [2], Reviewer rKoe pointed out that [1] cannot ensure constraint satisfaction because it uses a soft constraint approach.
> Thus, it is questionable, from our point of view, whether constraint violations can be prioritized over the objective function during gradient descent.
>
> In contrast, our reward function imposes an explicit priority ordering between the objective function and the constraints, with constraint violations taking precedence over objective values (see Eq. 4).
> Our theorem establishes feasibility by linking violations to the reward function, such that improvements in feasibility are directly aligned with obtaining non-negative rewards.
> By repeatedly following reward-maximizing behavior, the agent reaches a point where feasibility can no longer be improved, thereby guaranteeing that a feasible solution is found.
> Figure 10 of our revised paper shows that RL-SPH gradually reduces the number of violated constraints and eventually satisfies all of them, thereby discovering the first feasible solution.
> This empirical evidence further supports our claim.
>
> Since the loss function in [1] follows a soft constraint approach that allows constraint violations, it may lead to a situation where the loss for objective values dominates the loss for constraint violations during gradient descent, due to the absence of an explicit priority ordering.
> Given that the theorems in [1] do not explicitly address the objective function and its priority, this casts doubt on whether the feasibility guarantee has been comprehensively justified.
> Accordingly, it appears that the paper should be subjected to a rigorous peer-review process before reaching any definitive conclusions.
>
> Thus, we would like to clarify our contribution as follows: *To the best of our knowledge, RL-SPH is the first E2EPH that provides a theoretical feasibility guarantee for ILP (Proposition 1).*
> We remain open to revising this statement if you have any further suggestions.
>
> [1] Learning to Optimize for Mixed-Integer Non-linear Programming with Feasibility Guarantees. arXiv, 2024.
>
> [2] https://openreview.net/forum?id=1oIXRWK2WO, OpenReview ICLR, 2025.
>
> # A.7 Random initialization on NBI
> > "Was random initialization more difficult or unstable in this setting (e.g., due to unbounded variable ranges)?"
>
> We examine whether RL-SPH can still find feasible solutions for NBI with the random initialization.
> As shown in the below table, RL-SPH with random initialization achieves a 100\% FR, indicating that RL-SPH can be trained to obtain feasible solutions from the randomly generated initial point even on NBI.
> However, the solution quality obtained from random initialization is significantly worse than that from LP initialization.
> We conjecture that this stems from the unbounded range of the integer variables, which allows them to take larger values.
> The observation that Random(avg), which assigns values greater than 1, yields lower PG and PI than Random(1) supports this explanation.
> This gap in solution quality is therefore attributable to the initialization strategy, underscoring the need for future research on initialization methods that are more tailored to general ILPs while faster than LP.
>
> | Dataset | Initialization type | FR(%)↑ | PG(%)↓ | PI↓ |
> |:-------:|:-------------------:|:----------:|:-------------:|:------------:|
> | | LP | 100 | 0.00±0.00 | 10.8±0.8 |
> | NBI | Random(1) | 100 | 30.15±3.38 | 19.8±1.3 |
> | | Random(avg) | 100 | 27.79±2.17 | 17.6±0.9 |
> - The performance of RL-SPH on NBI under different initializations. Random(1) assigns the value 1 to randomly selected variables, identical to the method described in Appendix C. Random(avg) assigns the average of the non-zero values from the previous instance; apart from that, it works the same as Random(1). During training, the average value was calculated as 4, and Random(avg) uses it for assignment in the test phase.

---

### Author Response · Authors · 2025-12-02
**Final Remarks by Authors**

Dear Reviewers and Area Chairs,

We propose a novel RL–based start primal heuristic, RL-SPH, which learns to generate high-quality feasible solutions for ILP without reliance on external methods.
RL-SPH addresses a more challenging problem than existing E2EPH methods, as it must generate feasible solutions without relying on an external solver.
To the best of our knowledge, RL-SPH is the first E2EPH that provides a theoretical feasibility guarantee for ILP.
We also prove that RL-SPH attains a feasible solution for ILP within polynomial time when the condition $\mathcal{T} \le n$ is satisfied.
Advancing learning-based methods for achieving feasibility establishes a crucial foundation for ML-based stand-alone solvers pursuing optimality, especially given that achieving optimality is recognized as a major challenge in the ML for CO community [1].
To further clarify the novelty and contributions of our work, we have added an additional explanatory section in Appendix G.

Experiments on five benchmarks show that RL-SPH achieved a 100% feasibility rate (FR), even for non-binary ILP.
RL-SPH also attained a 44× lower primal gap and 2.3× lower primal integral on average compared with classical start primal heuristics.
To enhance the paper’s overall impact, we also conducted a comparison with DiffILO [2], the state-of-the-art E2EPH method.
While DiffILO exhibits a 0% FR on two of the three datasets, RL-SPH consistently achieves a 100% FR across all datasets.
Furthermore, RL-SPH requires on average 34× less training time than DiffILO.

In accordance with the reviewers’ requests, the work we conducted during the discussion period is summarized as follows:
- **Experiments.** We have incorporated the following experiments into the manuscript:
    - Joint model on MIPLIB instances having integers (Table 4 in Section 4.4)
    - Comparison with SOTA E2EPH (Appendix J.1)
    - Random initialization on NBI (Appendix J.4)
    - Effect of the neighbor variable ratio (Appendix J.5.1)
    - Effect of the toward-optimal bias (Appendix J.5.2)
    - Behavior of the two-phase reward system (Appendix J.6.1)
    - Long-range variable dependencies (Appendix J.6.2)
    - Training curves (Appendix J.7)
- **Paper revision**. We have revised the manuscript with respect to the following points:
    - Ensuring the condition of our polynomial-time guarantee (Proposition 2)
    - Details on the computational cost of Algorithm1 (Appendix B)
    - Clarification on illustration and description of our GNN (Section 3.2)
    - Clarification on using the same search routine for both training and testing (Section 3.3)
    - Clarification on stopping criteria (Section 4.1.2)
    - Clarification on wall-clock time reporting (Section 4.1.3)
    - Addition of a new wall-clock time metric (Section 4.1.3)
    - Addition of test-time solving curves (Figure 5 in Section 4.2)
    - Clarification on failure handling in RL-SPH (Appendix J.3)
    - Reorganization of several sections (see A.1.2 of our response to Reviewer EHiR)
    - A brief summary of the appendix at the end of the main body
    - Addition of a table of contents for the appendix
- **Responses to remaining comments.** We addressed the reviewers’ comments on the following topics; readers may navigate to any topic by searching for its corresponding title on this page. Some of these have also been incorporated into the manuscript:
    - Positioning relative to prior work (Appendix G)
    - Clear rationale on reward design (Appendix H.1)
    - Novelty of the feasibility-aware search strategy (Appendix K.5)
    - Trinity as the key enabler (Appendix K.6)
    - Potential bottlenecks at extreme scales (Appendix K.7)
    - Comparison of feasibility guarantees
    - Handling unbounded integers
    - Computational cost
    - No loop in phase1
    - Action magnitude
    - Handling extremely large scale ILP
    - Improving the incumbent value without harming feasibility
    - Comparison with other heuristics
    - Started with the basic Actor-Critic algorithm

We are sincerely grateful for your interest in our work and for the time you have invested in reviewing it.

Best regards,

The Authors

---
[1] Son et al. Meta-sage: Scale meta-learning scheduled adaptation with guided exploration for mitigating scale shift on combinatorial optimization. ICML, 2023.

[2] Geng et al., Differentiable Integer Linear Programming. ICLR 2025.

---

### Meta-Review · Area_Chair_qm7q · 2026-01-09

**Summary:**

The paper introduces RL-SPH, a novel Reinforcement Learning-based Start Primal Heuristic designed to independently generate feasible solutions for Integer Linear Programs (ILP). RL-SPH includes three novel designs: a Transformer-based Graph Neural Network, a feasibility-aware search strategy, and a two-phase reward. Empirically, RL-SPH achieves a 100% feasibility rate and higher solution quality across multiple benchmarks. During the rebuttal, the authors have addressed most concerns from reviewers. However, several concerns remain only partially resolved. Therefore, I recommend rejection.

**Reviewer Concerns:**

The rebuttal and revision addressed several major reviewer requests, including: (1) adding comparisons/positioning against recent learning-based end-to-end primal heuristics (Hj6o, m7aJ, EHiR), (2) providing additional ablations to support component attribution (pNKT, EHiR), and (3) clarifying aspects of the method and experimental protocol (pNKT, EHiR, Hj6o). However, the following concerns remain: (1) the polynomial-time guarantee is still conditional, and its assumption Τ≤n needs evidence to be true (m7aJ Weakness #4); (2) the latter added comparisons are still incomplete compared to the reviewers’ requests (Hj6o Weakness #1, m7aJ Weakness #3).

**Reviewer Scores:**

Reviewer EHiR have actively participated in discussion, so this reviewer’s score (borderline accept) is very likely to be the final decision. Reviewer pNKT rated borderline accept at the beginning with medium confidence. The concerns of pNKT were mostly addressed, but the response to Questions #2 seems not fully convincing, so this reviewer may keep the score. The remaining two reviewers were initially borderline reject and, while the rebuttal meaningfully improved the paper, some concerns are not fully addressed, so I think they tend to keep their scores after rebuttal.

---

### Decision · Program_Chairs · 2026-01-26

Reject